# Microphysical variability of Amazonian deep convective cores observed by CloudSat and simulated by a multi-scale modeling framework

J. Brant Dodson[1], Patrick C. Taylor[2], and Mark Branson[3]

[1]Science Systems and Applications, Inc., Hampton, VA, USA
[2]Climate Science Branch, NASA Langley Research Center, Hampton, VA, USA
[3]Department of Atmospheric Science, Colorado State University, Ft. Collins, CO, USA

*Correspondence to:* J. Brant Dodson (jason.b.dodson@nasa.gov)

**Abstract.** Recently launched cloud-observing satellites provide information about the vertical structure of deep convection and its microphysical characteristics. In this study, CloudSat reflectivity data is stratified by cloud type, and the contoured frequency by
altitude diagrams reveal a double-arc structure in deep convective cores (DCCs) above 8 km. This suggests two distinct hydrometeor modes (snow versus hail/graupel) controlling variability in reflectivity profiles. The day-night contrast in the double-arcs is about four times larger than the wet-dry season contrast. Using QuickBeam, the vertical reflectivity structure of DCCs is analyzed in two versions of the Superparameterized Community Atmospheric Model (SP-CAM) with single-moment (no graupel) and double-moment (with graupel) microphysics. Double-moment microphysics shows better agreement with observed reflectivity
profiles; however, neither model variant captures the double-arc structure. Ultimately, the results show that simulating realistic DCC vertical structure and its variability requires accurate representation of ice microphysics, in particular the hail/graupel modes, though this alone is insufficient.

## 1 Introduction

As a driver of the hydrological cycle, the frequency and intensity of atmospheric deep convection influences spatial and temporal
characteristics of precipitation. Our ability to simulate convective behavior on short (diurnal) and long (climate change) timescales significantly modifies projected changes in the spatiotemporal distribution of precipitation (including floods and drought), radiation, and other climate system elements (Arakawa, 1975). Simulating convection relies on our understanding of the physics controlling and modulating its behavior, including cloud microphysics, cloud scale dynamics (updraft/downdrafts), entrainment and other large-scale atmospheric interactions, cloud-surface interactions, and cloud-radiation interactions (Randall et al., 2003;
Arakawa, 2004).

Atmospheric convection exhibits variability on multiple time scales, including diurnal and seasonal. The convective diurnal cycle (CDC), a well-documented and important mode of variability, is particularly pronounced over land (Yang and Slingo 2001, Nesbitt and Zipser 2003, Tian et al. 2004, Kikuchi and Wang, 2008; Yamamoto et al., 2008). The CDC is characterized by a rapid insolation-driven transition from shallow to deep convection in the early afternoon, followed by either a slow decay through the
evening and early morning, or transition into mesoscale convective systems (MCSs), persisting into the next morning (Machado et al., 1998, Nesbitt and Zipser 2003). Geostationary satellite observations, in particular, have provided a global view of the spatial complexity of the CDC (Yang and Slingo, 2001). However, most satellite data sets only observe convective cloud top properties using passively-sensed visible and infrared radiances and cannot sense the cloud interior. With passive sensors, it is difficult to clearly distinguish between deep convective cores (DCCs) and related anvils, the latter having a diurnal cycle offset from DCCs
by approximately three hours (e.g. Fu et al. 1990; Lin et al., 2000; Zhang et al., 2008).

A small number of satellites, such as CloudSat (Stephens et al., 2008), carry radars that penetrate cloud tops, sensing the interior of deep convection. Spaceborne radars allow the examination of deep convection invisible to most satellites, especially internal structure and microphysics. CloudSat carries a W-band radar which is specifically attuned to observe smaller cloud liquid and ice hydrometeors. Other satellites, such as Tropical Rainfall Measuring Mission (TRMM) (Kummerow et al.; 1998) and the Global Precipitation Measurement (GPM) mission (Hou et al., 2014) are attuned to larger precipitation-sized hydrometeors. A sizable body of literature describes observations of tropical convection using these satellites (e.g. Petersen and Rutledge, 2001; Schumacher et al., 2004; Nesbitt and Zipser, 2003; Zipser and Nesbitt, 2007; Liu and Zipser, 2015; Liu and Liu, 2016). In addition to radars, there exist lidars such as that carried by Cloud-Aerosol Lidar and Infrared Pathfinder Satellite Observation (CALIPSO) (Winker et al., 2009), which has been used to examine the properties of convection and anvils (e.g., Sassen et al., 2009; Riihimaki and McFarlane, 2010; Del Genio et al. 2012).

Spaceborne cloud radars, precipitation radars, and lidars offer complementary views of tropical convection. Cloud radars sample the higher altitudes and anvils of DCCs, as well as associated stratiform clouds, but are ineffective at lower altitudes where precipitation-sized hydrometeors attenuate the radar beam. Precipitation radars examine the lower- and mid-level structure of DCCs, but cannot see cloud tops and anvils. Lidars are sensitive to the tops of thick clouds and can measure their altitude with high precision; however, they are unable to penetrate the tops of DCCs and most anvils. In this study, we will focus on the high-level features of DCCs and anvils, where CloudSat is most useful.

CloudSat's polar orbit, crossing the equator during early afternoon and early morning, provides two views of the CDC, near the beginning and end of the mean precipitation diurnal cycle over land. This ability has not been well-exploited in the literature. Liu et al. (2008) represents one of the few analyses on observed day-night differences using CloudSat. They surveyed day-night contrasts between reflectivity profiles over both tropical land and ocean, finding that high reflectivity clouds occur more frequently at night than during the day at all altitudes except at cloud tops (13 km). The authors interpret this difference as a consequence of the CDC, in which the peak in deep convective frequency occurs after the 1330 LST CloudSat overpass time, while there are still frequent lingering MCSs during the 0130 LST overpass. However, previous efforts muddle the physical interpretation by mixing the frequency of both shallow and deep convection, the vertical convective reflectivity profile, and the properties of other cloud types. Resolving these issues, we examine the day-night contrast in the reflectivity profile of mature deep convection after stratifying by cloud-type. This methodology allows the separation of convective frequency from the individual reflectivity profile signatures of different cloud-types, including those generated at different times in the convective life cycle, creating a clearer view of the day-night contrast.

The Amazon basin is an ideal location for studies of the CDC for multiple reasons. First, the Amazon has a well-defined, high-amplitude continental CDC, peaking regularly in the mid-afternoon (Yang and Slingo, 2001). Amazonia has a prominent recurring propagating coastal squall line, and various secondary local effects related to orography and the Amazon river (Janowiak et al., 2005; Burleyson et al., 2016). However, aside from the aforementioned squall line, it lacks the major diurnally propagating signals observed in other continental convective regions (e.g., the central United States and southern China) (Wallace et al., 1975; Dai et al., 1999; Zhou et al., 2008) that make generalizing regional CDC studies difficult. Second, Amazonia has a well-defined wet and dry season, allowing a seasonal examination of the CDC including distinct meteorological forcing for convection: locally-forced (common in the dry season) versus non-locally forced (common in the wet season). Finally, the Amazonian CDC alters the top-of-atmosphere radiative diurnal cycle (Taylor 2014a,b, Dodson and Taylor 2016), representing an influence on regional and global climate that meteorological reanalyses and climate models struggle to simulate (Yang and Slingo, 2001; Itterly and Taylor, 2014; Itterly et al. 2016).

This paper documents and describes a detailed view of the diurnal variability of the convective vertical structure observed by CloudSat. One of the key methods to accomplish this is to separate the variability in convective frequency from the variability in radar reflectivity. This new perspective not only clarifies previous findings, but also reveals a unique, previously unreported double-arc structure in the average radar reflectivity profile of deep convection. This new finding relates to the ice microphysical structure relevant to convective dynamic and thermodynamic properties, including precipitation rate, downdraft and cold pool strength (affected by evaporation and sublimation of hydrometeors), latent heating vertical profile (and associated warming and drying of the convective environment), and detrained water mass (McCumber et al., 1991, Grabowski et al., 1999, Gilmore et al., 2004, Li et al., 2005). Based on our results, the simulation of deep convective characteristics, and the comparison between simulation and observations, benefits from a detailed representation of ice microphysics.

## 2 Data and Methodology

### 2.1 Observations

Cloud observations are taken from CloudSat, a cloud-observing member of the A-Train (Stephens et al., 2008), orbiting at 705 km altitude, 98° inclination, and equatorial crossing time of 1:30am/pm local time. The primary instrument is the Cloud Profiling Radar (CPR), a 94 GHz radar with a 1.1 km wide effective footprint and 480 m vertical resolution, oversampled to create a 240 m effective vertical resolution. CloudSat operated as designed from June 2006 to March 2011, until suffering a battery malfunction. This time period serves as the temporal data domain.

For cloud-type stratification, the CPR Cloud Mask and Radar Reflectivity fields from the 2B-GEOPROF product (Mace 2007) are used. The cloud-type stratification identifies 4 cloud types, DCCs, anvils (AVN), clouds attached contiguously with DCCs (CLD-D), and other clouds (CLD). DCCs are identified using a CPR-based methodology on a profile-by-profile basis (i.e. with no consideration to neighboring columns) according to three criteria from Dodson et al. (2013):

(1) must be at least 10 km tall,

(2) must have a continuous vertical region of reflectivity of at least -5 dBZ between 3 km and 8 km altitude, and

(3) the maximum reflectivity value at any altitude in the middle troposphere (3 km to 8 km) must be at least 0 dBZ.

The lower altitude bound in criterion 2 is raised to 5 km when heavy precipitation is detected (indicated by low surface reflectivity), accounting for attenuation effects. The latter two criteria restrict the set of profiles with deep cloud layers to those which likely contain active, vigorous DCCs only. These criteria are guided by the cloud definitions used in creating the 2B-CLDCLASS product (Wang and Sassen, 2001). When the data are stratified by these criteria, CloudSat observed 187,457 vertical profiles of DCCs in the Amazon over the time domain, with just less than half (92,071, or 49%) occurring during the daytime overpass.

Figure 1 shows the spatial domain of the analysis region, centered on northern and central South America (25°S – 0°S, 70°W – 50°W). In addition, a CloudSat overpass through the domain displays the associated cloud presence, morphologies, and radar reflectivity for a single overpass (Fig. 1a-c), displaying the identified cloud types in the bottom panel (Fig. 1d). This particular overpass provides an example of the stratification method, displaying a variety of deep convective cloud systems and associated anvils, including narrow single-cell updrafts to the north, and wider multi-celled convection to the south. Note, only a subset of the profiles within deep convective cloud systems are labelled DCCs – this is deliberate to include only the profiles which likely contain active convective updrafts.

## 2.2 Modeling

To investigate the ability of models to simulate the observed DCC vertical structure and the influence of microphysics, we use the Superparameterized Community Atmospheric Model (SP-CAM) (Khairoutdinov et al., 2005). SP-CAM is a multi-scale modeling framework (MMF) replacing the convective parameterization (among other things) of the Community Atmospheric Model (CAM) with a cloud-resolving model (the System for Atmospheric Modeling, described by Khairoutdinov and Randall (2003)), coupled within each GCM grid point. This is a study of opportunity, using data made available from past work, and so the time domain is limited to an Amazonian dry season in the early 21$^{st}$ century. Only data from 0200 and 1400 LST are included in this analysis, which are closest to the CloudSat overpass times of 0130 and 1330 LST. We use two versions of SP-CAM, employing single moment (SPV4) and double moment (SPV5) microphysics. Hydrometeor mixing ratios for cloud ice, cloud water, rain, snow, and graupel (double moment-only) taken from the cloud-resolving model (CRM) component of SP-CAM are used to simulate the associated 94 GHz reflectivity profile using the QuickBeam radar simulator (Haynes et al., 2007). The CRM is based on the System for Atmospheric Modeling, and is run in two dimensional mode, with 4 km horizontal spacing and approximately 100 m – 1 km vertical spacing (varying by altitude). The formulation of SPV5, and the differences between SPV4 and SPV5, are documented by Wang et al. (2011). The only major differences between SPV4 and SPV5 of direct relevance to deep convection are the microphysical and radiative parameterizations; we attribute primary differences between SPV4 and SPV5 to microphysics.

A major difference between the SPV4 and SPV5 microphysics is the treatment of precipitating hydrometeors. SPV4 has diagnostic variables for snow and graupel. The SPV4 microphysics scheme predicts only non-precipitating and precipitating hydrometeors, which are partitioned into frozen and liquid by temperature. Snow and graupel are diagnosed from frozen precipitating water. However, snow and graupel as distinct hydrometeor species do not play a role in the prognostic microphysical equations in SPV4. In contrast, snow and graupel are prognosed hydrometeor types in SPV5, and are treated as a distinct species in the prognostic microphysical equations. So in addition to the reflectivity field calculated from precipitating ice in SPV4, we examine the influence of partitioning the precipitating ice into diagnosed snow graupel on the simulated radar reflectivity field. This new variant of SPV4 is hereby referred to as SPG4. Note that there are no differences between SPV4 and SPG4 other than the reflectivity fields simulated by QuickBeam. This lets us distinguish between the effects of adding graupel to the hydrometeor species, and the effect of switching from single- to double-moment microphysics. Because the hydrometeor parameters in QuickBeam are similar for precipitating ice and snow, the main effect of the partitioning is the increased reflectivity from diagnosed graupel.

It is difficult to directly compare satellite-retrieved and model-simulated convective cloud ice (Waliser et al., 2009). Radar reflectivity serves as a substitute basis for comparison, where model reflectivity is computed with a radar simulator. However, ice phase microphysical properties of DCCs are a key component in determining the model-simulated reflectivity profile. This creates a challenge for interpreting simulated reflectivity, as it is difficult to detangle the influence that model microphysics has on the simulated reflectivity profile and its relationship with other properties of simulated convection (e.g. vertical updraft velocity). Observed reflectivity profiles are affected by multiple ice hydrometeor types, including both snow and graupel/hail. In order for models to realistically simulate DCC reflectivity profiles, and thus allow for robust statistical reflectivity model/observation intercomparison (in the vein of Liu et al. (2008)), the models must simulate both ice hydrometeors realistically (a function of the microphysics) and the relationships between the hydrometeors and other aspects of convection (e.g. vertical velocity). Therefore, it is difficult to strictly and simply attribute model-observational differences to specific aspects of the parameterizations. Nevertheless, it is still possible and useful to show the aggregate effects that the choice of microphysics have on the simulated reflectivity field, and so (at least partially) account for model-observation differences.

## 3 Properties of convection as observed by CloudSat

### 3.1 Mean cloud properties

Untangling the influence of convective frequency on the deep convective vertical reflectivity structure benefits from an examination of the mean CloudSat-observed cloud properties (Fig. 2). First, we will look at the frequency of all cloud types, and then subset the clouds into DCCs and anvils. Clouds occur in a layer between 1.5 km and 12 km, with small maxima in cloud occurrence frequency (COF) at 11.5 km and 2.5 km (Fig. 2a). The vertical profile of cloud top heights (Fig. 2b) shows four regions of interest – a primary maximum at 13 km a secondary maximum at 1.2 km, a broad enhanced frequency region between 2.5 and 6 km, and a small maximum at 7.5 km. The cloud base heights (Fig. 2c) show a large primary maximum at 1.2 km, a broad secondary maximum centered at 11 km, and a small local maximum at 5 km. These features are consistent with the identified tri-modal vertical cloud structure in convectively-active tropical regions (Johnson et al., 1999; Khairoutdinov et al., 2009), with shallow convective clouds, cumulus congestus, and DCCs with associated anvils comprising the bulk of Amazonian clouds. The DCC-anvil and shallow cumulus modes are more prominent in the data than the cumulus congestus mode; this might be related to the greater variability of top heights for congestus than the other cloud types, which are constrained by the level of neutral buoyancy (for DCCs-anvils) and the atmospheric boundary layer top (for shallow cumulus).

Mean COF vertical profiles differ significantly between day and night, differences at most altitudes have p-values (two-tailed t-test) $\ll 0.01$. High (low) level clouds are enhanced during night (day), and the opposite suppressed. In addition, high altitude clouds (above 12.5 km) are more frequent during day than night. The cloud top height profiles (Fig. 2b) show a contribution to high level cloud frequency from a daytime top height increase (i.e. taller DCCs, likely with overshooting tops), indicating that despite fewer daytime high clouds they are taller than those at night.

DCCs occur in 3% of CloudSat profiles over Amazonia (Fig. 2d), with mean top heights near 14 km (Fig. 2e). The tallest DCCs reach an altitude of 18 km, which penetrate the tropopause and likely contribute to stratosphere-troposphere interactions (Johnston and Solomon, 1979; Corti et al., 2008; Avery et al., 2017). DCCs are on average about 0.5 km taller during the day than night (also significant at $p \ll 0.01$). Anvil cloud frequency (Fig. 2g) peaks at 12 km, with anvil top heights (Fig. 2h) maximizing at 13.5 km (0.5 km lower than DCCs). Anvil bases occur in a broad layer between 5 km (by definition the lowest altitude) and 11 km, diminishing with height above (Fig. 2i). The 5 km lower limit of anvil bases (where 5 km is chosen as being near the freezing line) is evidentially an artificial limit imposed by the methodology, and there may be no clear distinction between anvil clouds and deeper free-tropospheric clouds in nature.

### 3.2 Deep convective reflectivity profiles

The frequency component of the data for various cloud types has been isolated and described, so now the vertical structure variability is open for examination. The mean vertical profile of reflectivity in DCCs over the Amazon, as well as the total variability, are presented as contoured frequency by altitude diagrams (CFADs), where the color shading represent the probability density function of reflectivity at each altitude (Fig. 3). Previous research (e.g. Bodas-Salcedo et al. 2008, Satoh et al. 2010, Nam and Quaas 2012) associates deep convection with a characteristic arc shape in the reflectivity CFAD, maximizing in the middle troposphere and decreasing at upper and lower altitudes. A similar shape is apparent in Fig. 3. Reflectivity is reduced near the surface from radar beam attenuation by raindrops (Sassen et al., 2007). The kink in the reflectivity profile at 5 km is a "dark band" marking enhanced beam attenuation from melting hydrometeors at the freezing level. Reflectivity in the higher cloud altitudes (above 7.5 km) decreases with height primarily through reduction in hydrometeor size – this is because, assuming Rayleigh

scattering and ignoring phase changes, hydrometeor size dominates reflectivity (proportional to the sixth power of diameter) (Battan, 1973). Large hydrometeors fall out of the updraft more rapidly than small hydrometeors, leading to vertical size sorting.

The CFAD associated with the DCC vertical profile displays an interesting feature above 8 km. While the CFAD follows the characteristic arc shape at lower altitudes, in the upper troposphere the CFAD splits into two arcs. The low-reflectivity arc decreases below 0 dBZ at 11 km, whereas the high-reflectivity arc remains above 0 dBZ at 14 km. The double-arc structure most likely indicates two different modes of hydrometeors: a low-reflectivity arc associated with snow and a high-reflectivity arc associated with graupel and hail. Cloud ice has typical reflectivity values below the minimum detection threshold of CloudSat (-28 dBZ), and does not contribute as much to the CFAD as the other ice hydrometeor species. This double-arc structure is not obvious (and thus unreported) in previous studies examining radar reflectivity profiles in deep convection (e.g. Bodas-Salcedo et al. 2008, Satoh et al. 2010, Nam and Quaas 2012) largely because the DCCs are not cleanly separated from other cloud types, leading to a blurred reflectivity structure.

How do we know that the double arcs are associated with different hydrometeor species? Figure 4b shows the reflectivity CFADs for anvil clouds. Instead of a double-arc reflectivity structure above 8 km, anvils have a single arc with reflectivity well below 0 dBZ above 10 km. The CFAD closely resembles the ones constructed by Yuan et al. (2011), specifically for thick anvils, where reflectivity values of 0 dBZ are frequent at altitudes of 8-10 km, and decreases rapidly with height. However, Yuan et al. show the reflectivity maximum extending 1-2 km higher in altitude than Fig. 5 does.

This single reflectivity arc corresponds with the low reflectivity (i.e. snow) arc observed in DCCs. This result is consistent with the hydrometeors in anvils consisting of the snow and cloud ice detrained from DCCs (modified by cloud processes as the anvils age), while dense ice hydrometeors either fail to be detrained into the anvil or quickly sediment from the anvil base immediately adjacent to the DCC. This corresponds with in situ measurements of anvil hydrometeors from West African convection (Bouniol et al. 2010). Note that some graupel particles are likely detrained into the anvils produced by DCCs with strong updrafts (Cetrone and Houze, 2009). However, these particles are not large and/or numerous enough to create the double-arc structure in the anvil reflectivity CFADs.

The presence of two distinct groups of hydrometeors in the upper cloud indicates a fundamental mode of variability in the DCC reflectivity profile. Higher (lower) reflectivity in the upper cloud indicate a larger (smaller) ratio of hail/graupel particles to snow. Dense, large, reflective particles generated in DCCs with higher vertical velocities are lofted higher into the upper cloud (Liu et al., 2007), linking the upper cloud reflectivity to updraft velocity. This relationship can be used as a proxy metric of convective intensity, and compared with other convective properties (e.g., frequency, top height, precipitation, radiative effects, etc). Liu et al. suggest that this metric may be more useful for characterizing convective intensity than cloud top height, a traditional convective metric.

### 3.3 Day Versus Night and Wet versus Dry Season Variability

The DCC reflectivity profiles, in particular the CFAD double-arc structure, show significant day-night and wet-dry season variability. The day-night contrast results show the upper cloud reflectivity is larger during day than night, by up to 4.5 dBZ at 12.5 km (Fig. 4j-l, solid line), which is caused by a more prominent high-reflectivity arc during the day than night. This feature also supports the conclusion that daytime updraft velocities are higher than nighttime velocities. This is consistent with the continental CDC, as described previously. Nighttime convection (midway between the afternoon peak and morning lull) is likely to be weakening and/or transitioning to MCSs (Machado et al., 1998), and exist in an environment partially contaminated by earlier convection and not being rejuvenated by insolation (Chaboureau et al., 2004).

These results clarify the day-night contrast presented by Liu et al. (2008). In CloudSat observations, DCCs are more frequent during nighttime than daytime. However, the DCCs occurring during the daytime overpass have larger updraft velocities than those at night. The positive day-night reflectivity difference in the upper cloud extends well below 12 km (the altitude indicated by Liu et al. (2008)), and represents a day-night contrast in the microphysical properties of the ice phase in DCCs. In summary, DCCs observed during the daytime overpass are less frequent, but taller, with larger vertical velocities, and more ice hydrometeors in the hail/graupel phases, than the DCCs observed at night.

The double-arc reflectivity structure in the upper troposphere exhibits seasonal differences. Amazonian convection exhibits strong seasonal variations in the frequency and other properties because of changes in forcing mechanisms (Fu et al., 1999; Marengo et al., 2001; Raia and Cavalcanti, 2008), also connected with variability in day-night contrasts. Figure 5 shows the day-night contrast in DCC reflectivity profiles during the wet and dry seasons. Season-specific results show the same qualitative pattern as the annual results – higher reflectivity during day (night) than night in the upper (lower) cloud. The amplitude of the difference is similar in both seasons, but the altitudes of enhanced daytime reflectivity is limited to above 10 km in the dry season. Overall, the day-night radar reflectivity contrast is four to five times larger than the wet-dry season contrast, underscoring day-night contrasts as a major mode of deep convective variability.

The CFADs show additional differences between wet and dry seasons. Wet season CFADs show a well-defined double arc reflectivity structure in the upper cloud, whereas the dry season CFADs do not, particularly at night. This result might be a consequence of the drier thermodynamical environment and aerosol characteristics of the dry season environment (including anthropogenic aerosol from biomass burning) (e.g. Andreae et al., 2004; Lin et al., 2006). However, it may also be a sampling artifact due to the smaller number of DCCs during the dry season (9,616) versus the wet season (78,034). To test the sample size influence, we implement a Monte Carlo-style random sampling methodology to reduce the wet season sample size to that of the dry season. Reducing the sample size of the wet season to that of the dry season obscures the double-arc reflectivity structure (not shown), so the influence of seasonality on the double-arc structure cannot clearly be attributed to seasonal changes in the convective environment.

## 4 Comparison with simulated cloud from a multi-scale modeling framework

These findings raise questions about ongoing modelling studies that use simulated radar reflectivity as a metric for convective activity. The observed CFADs depict complex structure and variability in convective reflectivity. Can models replicate this? In this section, we will examine the ability of SP-CAM to replicate the properties of DCCs observed by CloudSat, in particular the double-arc reflectivity structure. In addition, because the model provides additional information about the simulated atmosphere that cannot be easily observed, such as vertical updraft velocity, we will look at the relationships between the radar reflectivity fields and other aspects of the simulated convection.

### 4.1 Simulated reflectivity and vertical velocity profiles

Figure 6 displays the CFADs of Amazonian DCCs for SPV4 and SPV5. Both versions produce reflectivity values above 5 km more than 10 dBZ lower than observed. Specifically, the observed graupel/hail branch of the reflectivity arc is missing in both model versions. SPV4 is particularly unrealistic, as the microphysics scheme does not represent graupel. SPG4 diagnoses graupel, but their effect on the reflectivity profile is minor. There is an enhancement of reflectivity of 2 dBZ at 6 km, and a 1-2 dBZ reduction of reflectivity elsewhere in the profile. This is a result of switching from nonprecipitating ice to snow in QuickBeam,

meaning that including diagnosed graupel enhances the reflectivity profile by no more than 4 dBZ. SPV5 microphysics predicts graupel, and the upper troposphere reflectivity is larger than SPG4, showing additional improvement in the switch in microphysical schemes. But SPV5 has a lesser (but still noticeable) disagreement with observations. No model variant reproduces the observed double-arc structure, suggesting a fundamental deficiency in representing the behaviour of large ice hydrometeors in convective updrafts.

Figure 7 shows that the convective updraft velocities in both SPV4 and SPV5 never exceed 5 ms$^{-1}$, which is unrealistically low for deep convection in the Amazon (Giangrade et al., 2016). This is likely a major contributor to the low reflectivity in the simulated reflectivity above 10 km, and is likely related to the coarse resolution of the CRM (Petch et al., 2002; Bryan et al., 2003; Khairoutdinov et al., 2009). However, vertical velocity is not the sole contributor to the size of the simulated reflectivity. Surprisingly, the disagreement between SPV4 and SPV5 does not directly correspond with a proportionally large change in simulated updraft velocity profile. Despite upper cloud reflectivity being higher in SPV5 than SPV4, mean updraft velocity in the upper cloud decreases (and turns positive in the lower troposphere). In addition, SPV4 DCCs have net negative velocity below 3 km. This may not seem like an intuitive result initially, and closer to the properties of stratiform precipitation. However, it likely represents the thermal "bubble" nature of atmospheric convection (Scorer and Ludlam, 1953; Batchelor, 1954; Carpenter et al., 1998; Sherwood et al., 2013; Morrison, 2017). The DCC-identification method favors columns with high reflectivity in the mid-to upper troposphere. These columns usually contain strong updrafts at the same altitudes, which is the convective updraft thermal. In contrast, near the surface, the high-buoyancy air has already been evacuated into the thermal aloft, leaving neutral or negatively buoyant air in the lower troposphere. The selection process is not perfect, and Fig. 7a and 7b shows that strong downdrafts are occasionally included in the set of DCC profiles. Nevertheless, the net sinking motion below 3 km in SPV4 is consistent with deep convection.

**4.2 Simulated reflectivity stratified by updraft velocity**

The argument we present relies critically on the relationship between convective updraft velocity, the graupel phase of microphysics, and radar reflectivity. These are difficult to unweave in the observations, because of lack of direct vertical velocity observations, and the limitations of microphysical retrievals (particularly in scenes with heavy precipitation for W-band radars (Mace et al., 2007)). However, it is possible to separate them in the simulation by stratifying vertical reflectivity and hydrometeor profiles by updraft velocity. In Figs. 8-11, the CRM-level vertical profiles associated with DCCs are conditionally sampled by the maximum positive vertical velocity (denoted hereafter as $W_{max}$) occurring in each profile. The probability density function (PDF) of $W_{max}$ is also displayed – note that the low $W_{max}$ in SPV4 and SPV5 are not confined to either daytime or nighttime. Fig. 8 shows that for SPV4 SPG4, SPV5, DCC cloud top height increases as $W_{max}$ increases (with regression slopes of 1.50, 1.66, and 1.35 km (m s$^{-1}$)$^{-1}$, respectively, when calculated between 0 m s$^{-1}$ and 2.5 m s$^{-1}$). Furthermore, the echo top heights of low reflectivity values in the upper cloud (e.g., -10 dBZ, indicated by dark green) increase with $W_{max}$ at similar rates as cloud top heights for SPV4, SPG4 and SPV5 (1.29, 1.28, and 1.64 km (m s$^{-1}$)$^{-1}$). However, the echo top heights of larger reflectivity values such as 0 dBZ (indicated by yellow) increases almost negligibly with $W_{max}$ for SPV4 (0.00 km (m s$^{-1}$)$^{-1}$), compared with SPG4 and SPV5 (0.68 and 1.79 km (m s$^{-1}$)$^{-1}$). And even with diagnosed graupel included, the slope of 0 dBZ echo top height in SPG4 is only a third that of SPV5. This result confirms that the lower reflectivity in SPV4 DCCs compared with SPV5 and observations is not caused *solely* by weaker updrafts, and the microphysics scheme plays a key role in upper tropospheric reflectivity.

There are two obvious simple possible causes for the lack of a double-arc CFAD structure in the CRM component of SP-CAM, beyond the lack of prognosed graupel in SPV4 and SPG4. First, note that there is no discontinuous "step function" jump in the 0 dBZ echo top height. The hypothetical presence of a discontinuous jump in echo top height as $W_{max}$ increases would cause a

double-arc structure in the simulated CFAD, similar to observations. In other words, the PDF in reflectivity in the upper troposphere would be bimodal, with the low (high) reflectivity mode representing DCCs with low (high) $W_{max}$, Such a jump could arise, for example, if graupel forms at only large values of $W_{max}$, which would discontinuously boost DCC reflectivity at high $W_{max}$. This could also occur if hail was included in the microphysics. In this hypothetical case, the low $W_{max}$ in the CRM would cause the lack of the double-arc structure, because convective updraft velocity would rarely be large enough to cross the jump in echo top height at large $W_{max}$. Only the low reflectivity mode, i.e. the snow arc, of the CFAD would manifest. However, in the real case, this discontinuous jump does not exist in the CRM. If the discontinuous jump in reflectivity exists in reality, but the CRM fails to replicate it, then a missing double-arc structure in SP-CAM is not caused simply by weak updrafts in the CRM.

The second possibility for the discrepancy is that the real PDF for $W_{max}$ is bimodal, while the simulated PDFs have only one peak. Observed DCCs in the Amazon occur in three main organizational structures: afternoon disorganized "pop-up" convection, coastal squall lines, and basin-wide organized convection similar to oceanic mesoscale convective complexes (Tang et al., 2016). In addition, the Amazon has a wide range of aerosol environments, which may influence several properties of convection including updraft velocities (Andreae et al., 2004; Lin et al., 2006; Tao et al., 2012). These combination of effects may create a bimodal (or multi-modal) PDF of real $W_{max}$. While SP-CAM (in particular SPV5) can represent certain properties of organized convection, such as diurnal propagation (Kooperman et al., 2013), the CRM does not allow realistic organization of convection. This contributes to a unimodal PDF of $W_{max}$ which may be unrealistic. Until recently, the state of observations did not enable a robust analysis to test these two possibilities; large samples of DCC vertical velocity are difficult to collect. However, recent observations from field campaigns such as the Green Ocean Amazon experiment (Martin et al., 2016, 2017) may be useful for testing.

**4.3 Hydrometeor variability by updraft velocity**

How do the ice hydrometeor species contribute to radar reflectivity? Figures 9 and 10 show the change in snow water content (SWC) and graupel water content (GWC) with $W_{max}$, respectively. In the case of SPV4, precipitating ice is classified as SWC. SPV4, SPG4, and SPV5 have SWC increasing at all altitudes above 5 km as $W_{max}$ increases (146, 99, and 166 mg (m s$^{-1}$)$^{-1}$, respectively), though SPV5 increases somewhat more rapidly than the others. Because both models show similar relationships between SWC and $W_{max}$, SWC alone cannot explain the reflectivity differences. The result shown by Fig. 10 can be summarized in two key points. First, SPV5 produces graupel at all values of $W_{max}$; therefore, the difference in reflectivity CFADs between SPV5 and the observations cannot be explained by the assumption that SPV5 simply has convective updrafts too weak to produce graupel. Second, the sensitivity of GWC to $W_{max}$ is slightly greater than that of SWC (183 versus 166 mg (m s$^{-1}$)$^{-1}$, respectively). This relatively rapid increase of GWC with $W_{max}$ in SPV5 is the best explanation for the rise of the 0 dBZ echo top height in SPV5, which is missing in SPV4. The increase in graupel with $W_{max}$ in SPG4 (55 mg (m s$^{-1}$)$^{-1}$) is much lower than that of SPV5, contributing to the lower radar reflectivity in SPG4.

Is the error in radar reflectivity for SPV4 related to changes in the convective updraft dynamics? Figure 11 shows vertical velocity profiles sorted by $W_{max}$ for SPV4 and SPV5 (note that vertical velocity in SPV4 and SPG4 are identical). The vertical structure for both SPV4 and SPV5 relate to $W_{max}$ in similar manners, with relatively strong ascent above 5 km to the cloud tops for most values of $W_{max}$, and neutral to weak downdrafts below 5 km. It does not appear that the differences in radar reflectivity between SPV4 and SPV5 are related to differences in the vertical velocity profile.

Improving the microphysical parameterization in CRMs, including both adding ice-phase hydrometeor species and shifting from single- to double-moment microphysics, results in noticeable improvements to the radar reflectivity fields associated with deep convection. This is consistent with previous studies which found improvements in the representation of deep convection as a result of the switch to multi-moment microphysics (e.g., Swann, 1998; Morrison et al., 2009; Dawson et al., 2010; Van Weverberg et al.,

2012; Igel et al., 2015). However, it is also clear that there are other reasons for the low reflectivity in the upper troposphere, such as the weak updrafts in both versions of the models (which should increase with improved model dynamics and resolution). In addition, it appears insufficient to capture observed variability in the DCC vertical structure, such as the double-arc reflectivity structure. Further improvements in microphysics will likely be necessary for CRMs to produce the full observed variability in the reflectivity field.

## 5 Summary and Discussion

We have presented an analysis of the DCC vertical structure in the Amazon observed by CloudSat. While the vertical reflectivity structure of convectively-active areas has been examined previously, the methodologies mixed together the vertical structure of deep convection, the frequency of deep convection, and the attributes of other cloud types. To clarify, we separate vertical profiles from DCCs and examine the variability in the vertical structure. The results reveal a distinctive double-arc structure in the CFAD related to the relative frequencies of snow and graupel/hail in the upper cloud, depicting variability in microphysics and updraft velocities. The graupel/hail branch of the double arc is more prominent during early afternoon than early morning, indicated by higher upper cloud reflectivity during day than night. This indicates stronger updrafts in mature DCCs during day than night. The day-night contrast in reflectivity structure is roughly four times larger than the contrast between the wet season and dry season, indicating that the day-night contrast is a prominent mode of DCC variability.

The results show the importance of separating data by cloud type before interpretation. This allows for a clearer process-based analysis of satellite observations, rather than a statistical view that mixes meteorological processes reducing their utility for aiding model improvement and process-level understanding. Future research should display caution when directly comparing the statistics of observed reflectivity and simulated reflectivity, and drawing conclusions about the accuracy of simulated convection from reflectivity statistics alone.

In addition, our results indicate that cloud resolving and related models, such as MMFs, are unable to capture the previously unreported double-arc structure, representing a weakness in the simulation of convection and is at least in part due to ice microphysics. The model-data comparisons suggest significant model deficiencies in the representation of radar reflectivity associated with convection remain, however more sophisticated model physics (e.g. switching from single moment to double moment microphysics, including more ice hydrometeor species) can significantly improve the representation. Until that time, care should be taken when using radar simulators to compare models with observations, especially when variability in the reflectivity field is being examined. These findings aid us in interpreting the relationship between radar reflectivity and convection, and particularly when comparing CloudSat observations with simulated reflectivity profiles in cloud resolving models.

*Data availability.* CloudSat data are available at the CloudSat Data Processing Center of Cooperative Institute of Research in the Atmosphere (http://www.cloudsat.cira.colostate.edu). SP-CAM data were provided upon request from the Center for Multi-Scale Modeling of Atmospheric Processes (http://www.cmmap.org).

*Competing interests.* The authors declare that they have no conflict of interest.

*Acknowledgements.* This work has been supported by NASA grant # NNH13ZDA001N-TERAQ, "The Science of Terra and Aqua"; by the National Science Foundation Science and Technology Center for Multi-Scale Modeling of Atmospheric Processes (CMMAP), managed by Colorado State University under cooperative agreement No. ATM-0425247; by Science Systems and Applications, Inc. under STARSSIII; and by the NASA Postdoctoral Program. The authors thank David A. Randall for providing assistance with using SP-CAM.

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

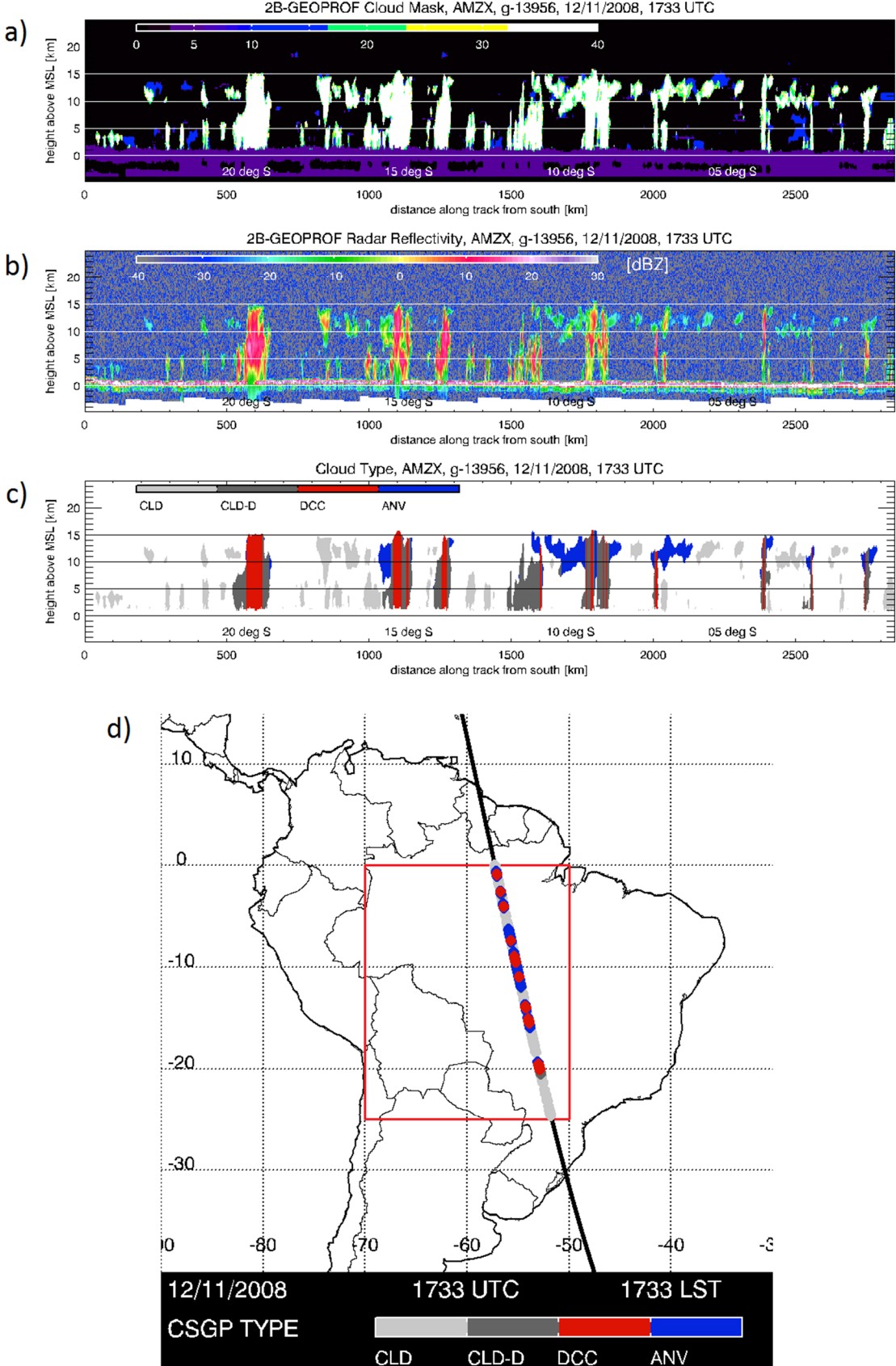

**Figure 1. (a-c) Example of a CloudSat cross-sectional observation of afternoon convection on 11 December 2008 at approx. 1733 UTC (1333 LST). Left-to-right on the x-axis corresponds with south-to-north. (a) is the cloud mask product from 2B-GEOPROF, with colors representing the cloud mask value corresponding with certainty of cloud identification. (b) is radar reflectivity, and (c) is cloud type.**

 Red indicates DCCs, anvils are indicated with blue, dark gray indicates clouds attached contiguously with DCCs, and light gray indicates other clouds.

(d) Map of northern South America with the study region (25°S – 0°S, 70°W – 50°W) marked with the red box. The heavy black line crossing the study region indicates the path of the CloudSat swath shown in the top panel. The dominant cloud type observed by CloudSat along the path is indicated by colored dots.

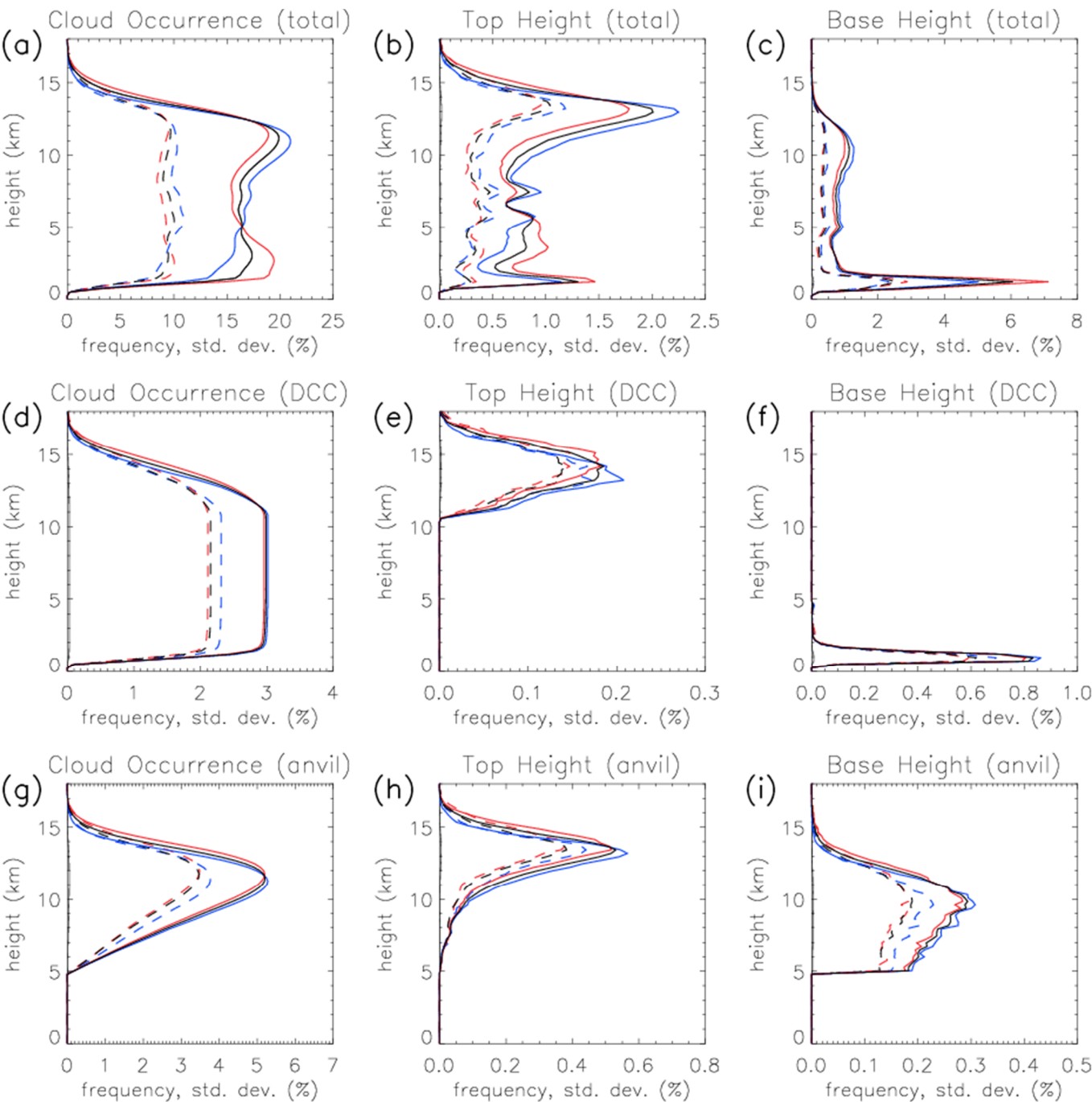

Figure 2. (Top) Vertical frequency profiles of (a) cloud occurrence, (b) cloud top heights (center), and (c) cloud base heights. Black lines are for day/night, and red (blue) is for day-only (night-only). Solid (dashed) line is the mean (standard deviation)

(Middle) Same as top, but for (d) DCC-only occurrence, (e) top heights, and (f) base heights.

(Bottom) Same as top, but for (g) anvil-only occurrence, (h) top heights, and (i) base heights.

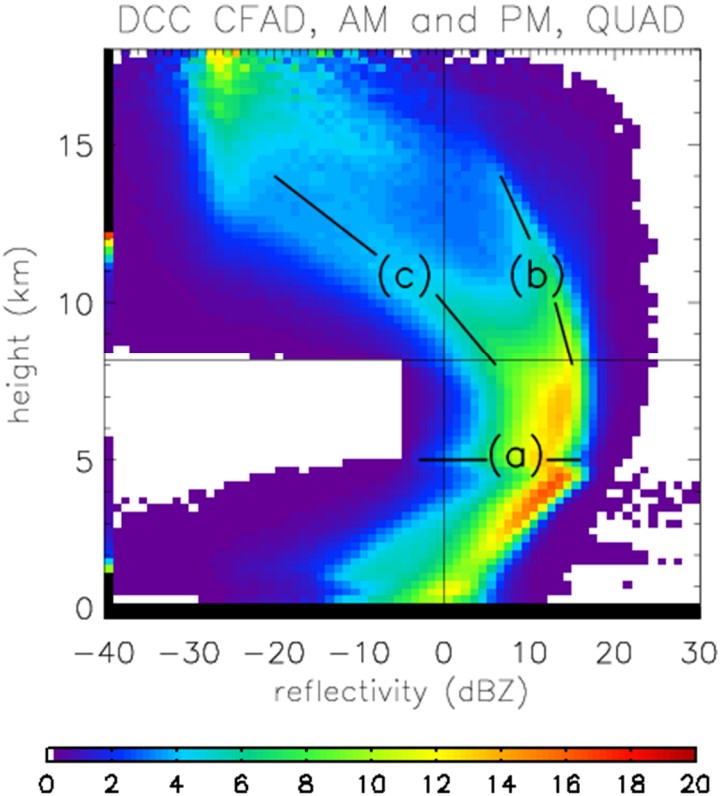


**Figure 3. The Contoured Frequency by Altitude Diagrams (CFADs) of reflectivity for DCCs in Amazonia. The colors represent the probability density function (PDF, as percentage) of radar reflectivity at each 240 m-tall layer observed by CloudSat. The vertical (horizontal) black line indicates 0 dBZ (8 km). Three distinct features of the CFAD, discussed in the text, are labelled on the figure. (a) indicates the dark band, with the horizontal line segments showing the mean altitude. (b) and (c) mark the locations of the high and low reflectivity arcs, respectively. The diagonal line segments show the orientation of each arc, and the rough values of the PDF modes.**


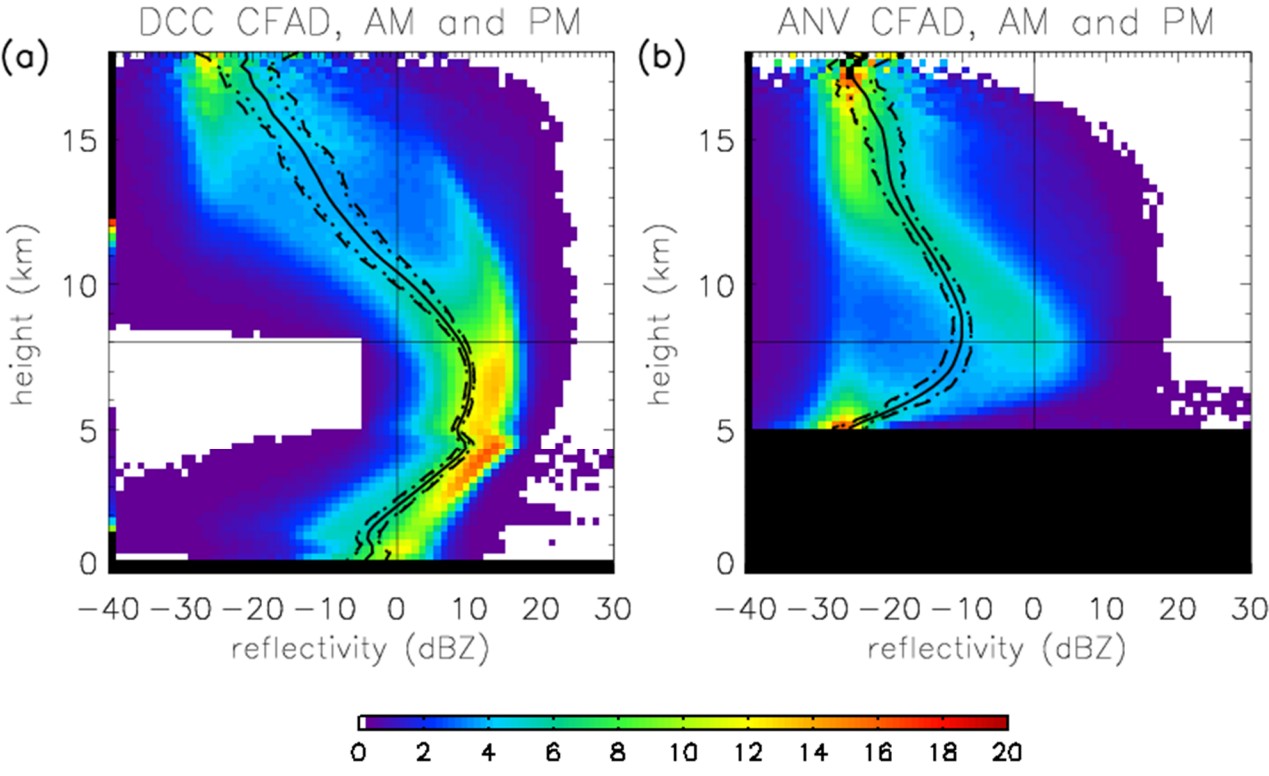

**Figure 4. (a) The CFAD for DCCs, same as Fig. 3. The black curves near the center of the data are the average reflectivity profiles. Dashed lines are the standard deviation bounds. (b) Same as (a), but for anvils.**


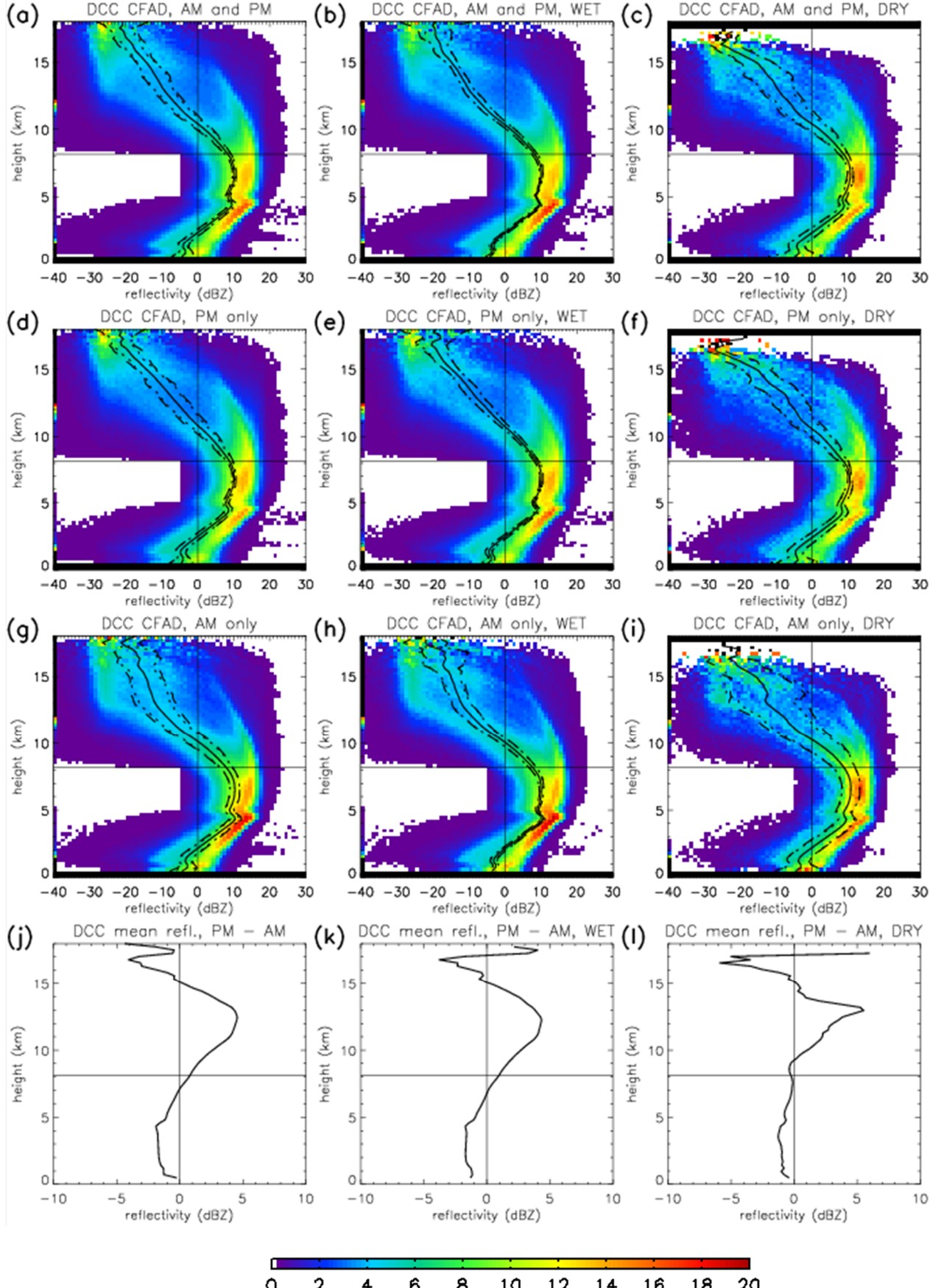

Figure 5. (a-i) The CFADs of reflectivity for DCCs in Amazonia separated by time of day and season. The left column is for all four seasons (wet, dry, wet-to-dry, and dry-to-wet; the middle column is for the wet season (WET); and the right column is the dry season (DRY). The first row are results for both times of day, the second row is day-only, and the third row is night-only.

(j-l) The difference between the day and night mean reflectivity profile values – i.e, the second row minus the third row.

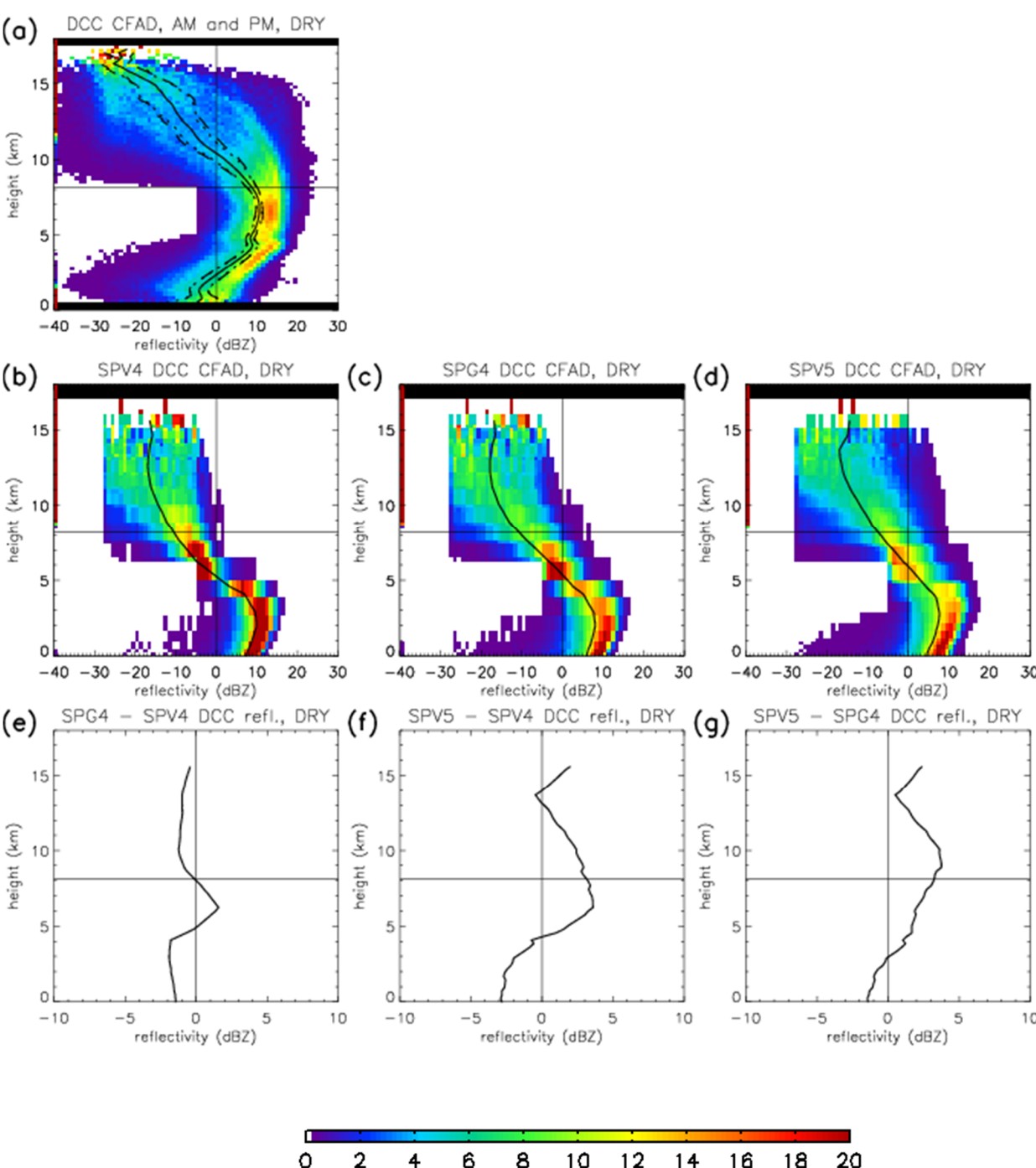

Figure 6. (a,b,c,d) CFADs of dry season DCC reflectivity in Amazonia from (a) CloudSat, (b) SPV4, (c) SPG4 (i.e. SPV4 with diagnosed graupel included), and (d) SPV5, as in Fig. 4.

(e,f,g) The difference of the mean vertical reflectivity profile between (f) SPG4 and SPV4, (f) SPV5 and SPV4, and (g) SPV5 and SPG4.

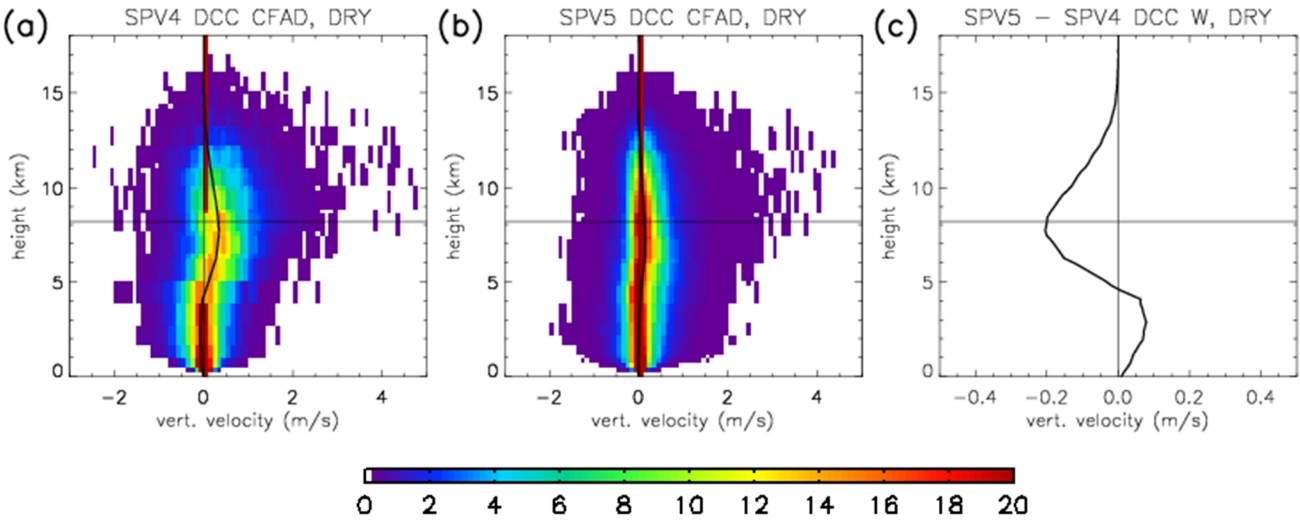

**Figure 7. (a,b)** CFADs of DCC vertical updraft velocity from **(a)** SPV4 and **(b)** SPV5. The format is the same as for the reflectivity CFADs. Note that SPV4 and SPG4 vertical velocities (not shown) are identical. **(c)** The difference between SPV4 and SPV5 mean vertical velocity profiles.

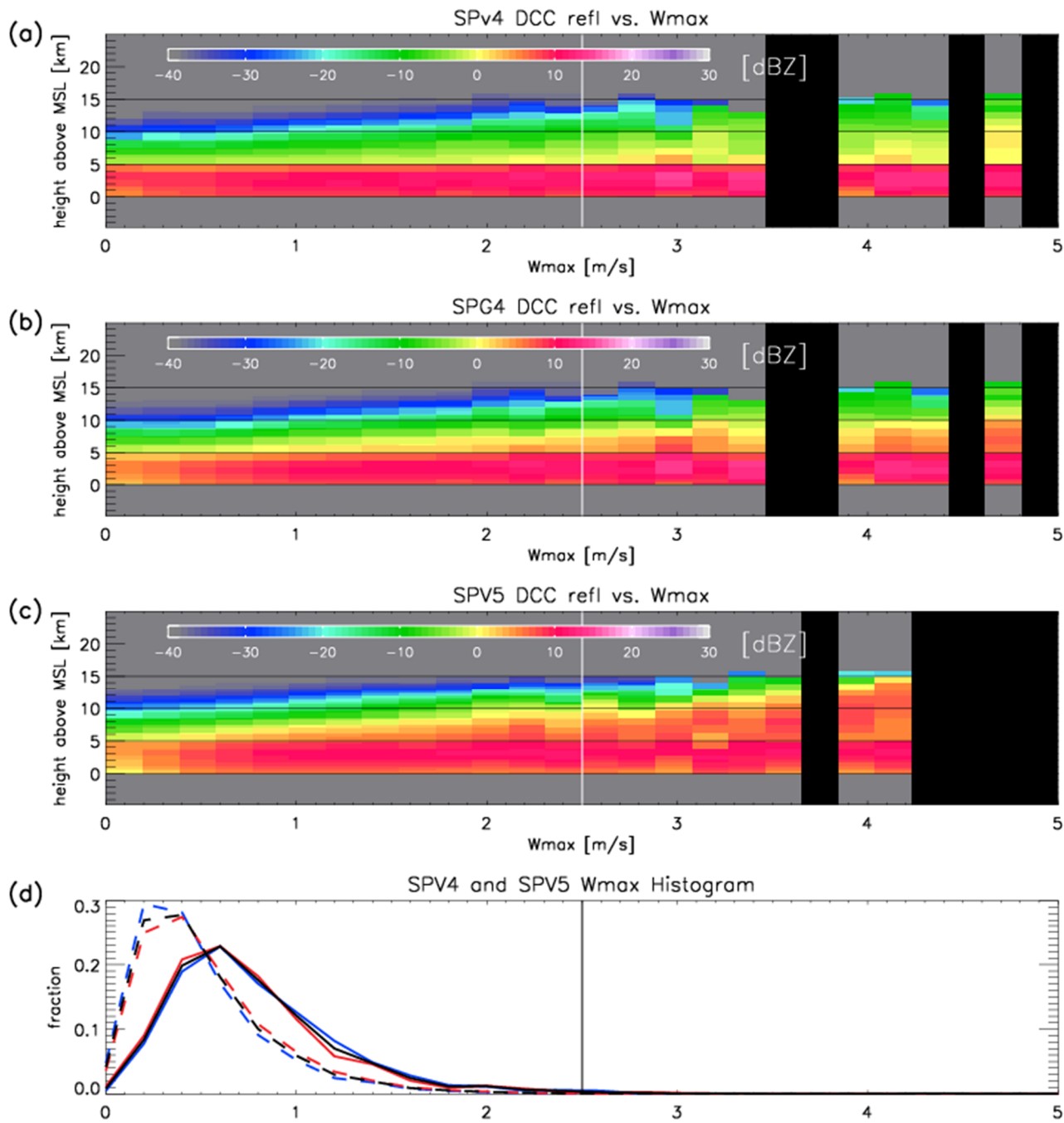

**Figure 8. (a) Mean vertical profiles of simulated DCC radar reflectivity in SPV4 sorted by maximum updraft velocity. The vertical lines at 2.5 m s$^{-1}$ represent the maximum cut off W$_{max}$ value for statistical calculations. Vertical black stripes are updraft values with no DCC occurrences.**

**(b) Same as (a), but for SPG4 data.**

**(c) Same as (a), but for SPV5 data.**

**(d) PDFs of maximum DCC updraft velocity for (solid) SPV4 and (dashed) SPV5. The black line represents data from both 0200 and 1400 LST, the blue line from 0200 LST only (night), and the red line from 1400 LST only (day).**

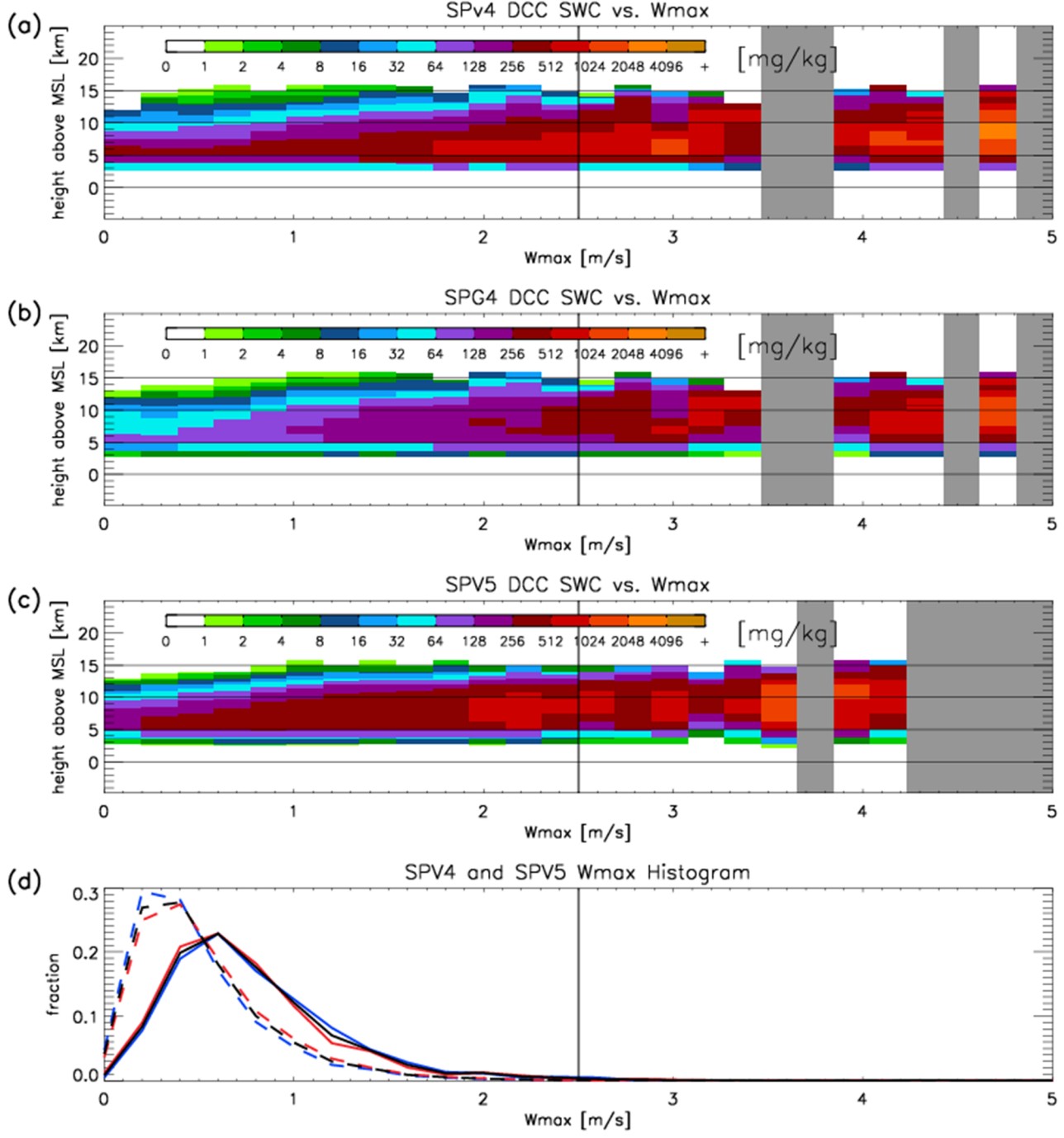

**Figure 9. Same as Fig. 8, but for snow water content (SWC). Because SPV4 does not distinguish between snow and graupel, all precipitating ice are depicted as SWC.**

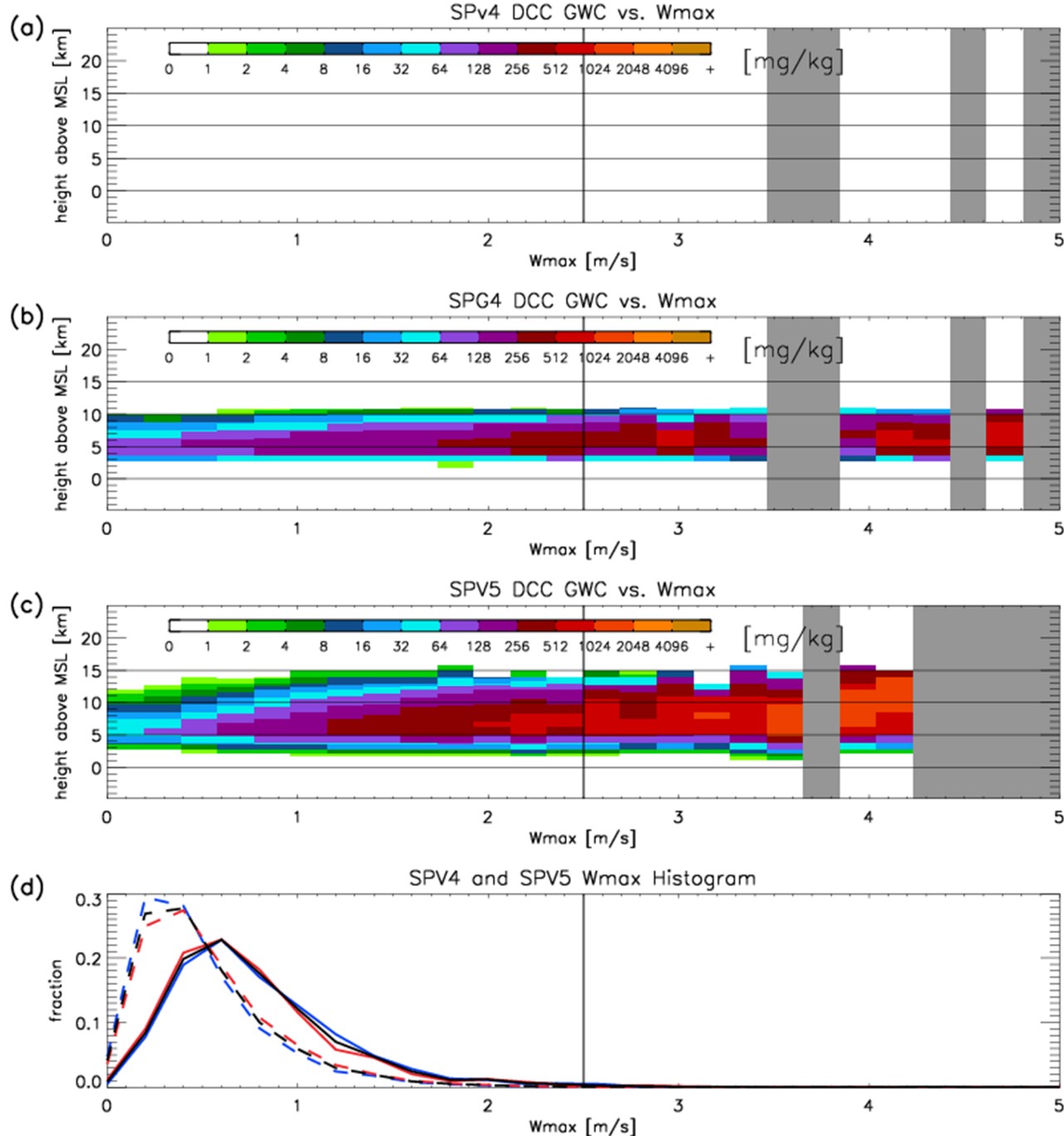

610

**Figure 10. Same as Fig. 8, but for graupel water content (GWC). Note the absence of GWC for SPV4.**

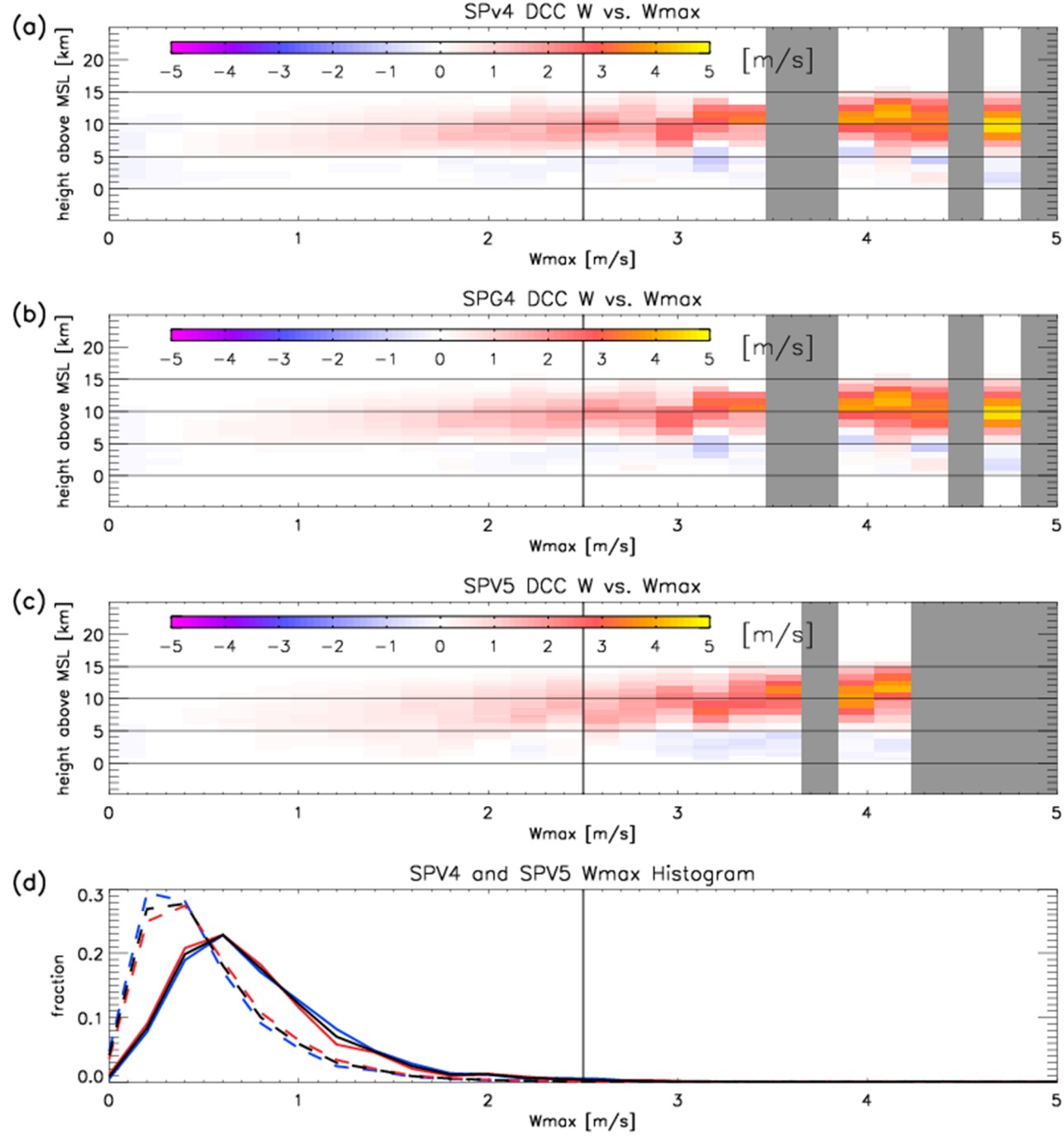

**Figure 11. Same as Fig. 8, but for DCC vertical velocity profiles. Note that the results for SPV4 and SPG4 are identical.**