# Peer review of "Microphysical variability of Amazonian deep convective cores observed by CloudSat and simulated by a multi-scale modeling framework"

_Atmospheric Chemistry and Physics, 2017_

## Referee Comment (RC1) · Anonymous Referee #1 · 6 Nov 2017

Please see the supplement for the complete review. The summary and major items are included below.

Summary: The authors use the Amazon as a testbed for assessing the internal structure of deep convection observed by CloudSat. Deep convective cores are shown, through a "double arc" structure in CFADS, to be composed of either highly reflective graupel and hail or weakly reflective snow. Cloud structure is contrasted between day/night and wet season/dry season to modest effect. The authors then compare their CloudSat results with those from two SP-CAM runs. These simulations are conducted with different versions of the model which results in the simulations of differing

[Figure]

cloud structure between the simulations themselves and between the simulations and CloudSat. The authors report new results, but these are incremental. There are several aspects of the paper that need improvement: 1) the "double arc" is not plainly obvious yet the authors make a point of discussing it at length; 2) the analysis of the simulations seems to lack an obvious direction. I would recommend acceptance if the issues below are addressed.

Primary items:

1) The "double arc" is not especially obvious in any of the panels of Fig. 3. It took me quite a while to fully recognize what structure the authors were talking about and to convince myself that it was not just a result of the contour intervals used. I'm not sure what the remedy is for this, but the double arc structure needs to be made clearer through either some enhancement of the figure, a schematic, or particularly lucid writing.

2) I don't understand why the authors feel they can ignore graupel in SPV4. The model seems to include graupel to the same degree that it includes any physical species. It seems to be just as much a part of the precipitating ice category as snow.

3) I don't think you have shown sufficient evidence to draw the conclusion you do on Line 285 (even if we all hope that this conclusion is true). Figure 6-9 show only that SPV5 behaves more logically. We do not know how the real world binned variables (reflectivity, SWC, etc) depend on Wmax. And, I'm not sure I agree that the SPV5 CFAD is more like the CloudSat CFAD than the SPV4 CFAD; they share more characteristics with each other than they do with CloudSat. Perhaps you could add the difference between the CloudSat CFAD mean and those from both model runs to Fig. 5d. Or maybe you could compare the variance at each level.

Please also note the supplement to this comment:
https://www.atmos-chem-phys-discuss.net/acp-2017-864/acp-2017-864-RC1-

supplement.pdf

[Figure]

**Supplement:**

"Microphysical variability of Amazonian deep convective cores observed by CloudSat and simulated by a multi-scale modeling framework" by Dodson, Taylor, and Branson.

Summary:

The authors use the Amazon as a testbed for assessing the internal structure of deep convection observed by CloudSat. Deep convective cores are shown, through a "double arc" structure in CFADS, to be composed of either highly reflective graupel and hail or weakly reflective snow. Cloud structure is contrasted between day/night and wet season/dry season to modest effect. The authors then compare their CloudSat results with those from two SP-CAM runs. These simulations are conducted with different versions of the model which results in the simulations of differing cloud structure between the simulations themselves and between the simulations and CloudSat. The authors report new results, but these are incremental. There are several aspects of the paper that need improvement: 1) the "double arc" is not plainly obvious yet the authors make a point of discussing it at length; 2) the analysis of the simulations seems to lack an obvious direction. I would recommend acceptance if the issues below are addressed.

Primary items:

1) The "double arc" is not especially obvious in any of the panels of Fig. 3. It took me quite a while to fully recognize what structure the authors were talking about and to convince myself that it was not just a result of the contour intervals used. I'm not sure what the remedy is for this, but the double arc structure needs to be made clearer through either some enhancement of the figure, a schematic, or particularly lucid writing.
2) I don't understand why the authors feel they can ignore graupel in SPV4. The model seems to include graupel to the same degree that it includes any physical species. It seems to be just as much a part of the precipitating ice category as snow.
3) I don't think you have shown sufficient evidence to draw the conclusion you do on Line 285 (even if we all hope that this conclusion is true). Figure 6-9 show only that SPV5 behaves more logically. We do not know how the real world binned variables (reflectivity, SWC, etc) depend on $W_{max}$. And, I'm not sure I agree that the SPV5 CFAD is more like the CloudSat CFAD than the SPV4 CFAD; they share more characteristics with each other than they do with CloudSat. Perhaps you could add the difference between the CloudSat CFAD mean and those from both model runs to Fig. 5d. Or maybe you could compare the variance at each level.

Other items:

Line 34: But Fu et al (1990) uses passive sensors. It should be made clear that the second two citations use models.

Line 42: "There exist lidars" is awkward phrasing.

Line 57 to 63: These lines describe the most significant impact this paper will make several pages later.

Line 88: You might want to rephrase "temporal data domain".

Line 107: Up to this point, you have not explained what scientific purpose you have in focusing on DCC. Is there a reason we expect Amazonian DCC to be especially microphysically variable?

Line 110: You may want to make it clearer that Fig. 2 does not use your DCC selection.

Line 122-123: These two sentences seem contradictory.

Figure 3a: What does "Quad" signify?

Line 150: How do you know that neglecting to separate the data is the reason the double arc has not shown up before? It seems like you could show your selected-CFAD and an all data-CFAD.

Line 167: I'm not sure what the author is implying through the end of this paragraph.

Figure 3j: The standard deviation of what? Or is it the difference of the standard deviations?

Line 192: Why not show Wet-Dry differences in an additional column (like PM-AM differences are shown in an additional row)? Well, OK, the authors answer this question by the end of the paragraph by punting on seasonal differences. But if they want to do this (which they can), they should probably change the title of section 3.3 to just "Day versus night variability".

Line 220: How was the CRM sampled? Did you sample individual CRM grid columns in a way similar to CloudSat's sampling (i.e. approximately north-south at 1:30am/pm)?

Line 226: I'm confused about what the authors mean throughout this paragraph. In SPV4, according to Wang et al (2011), SAM diagnoses all its microphysical species from predicted precipitating and non-precipitating water. Why do the authors feel they can "disregard diagnosed graupel" (and by that I assume they mean when running the radar simulator) and not, say, snow which seems to be diagnosed in the same manner as graupel? It seems like the authors should include graupel in their calculations and change their conclusion that SPV4 underestimates graupel. This would also make the paragraph beginning on Line 230 unnecessary.

Line 238: Except SPV4 does "represent" graupel.

Figure 5: why use the dry season when the wet season produces the more significant "double-arc"? The vertical velocities seem oddly low in magnitude. There is barely any weight to the PDFs above 1m/s (which would usually be considered "convective" in a CRM).

Figure 6: Realistically, I think one has to question the sampling at high $W_{max}$ when there are intermediate bins with zero samples. All the panels should probably be cutoff at ~3m/s. What are the different colored lines in panel (c)? Also, the label on the y-axis should probably be something like "fraction" rather than "count".

Line 264: This discussion of the "jump" needs to be clearer.

Line 270: Or this logical thread about how the double arc structure might arise isn't correct. I don't have a different hypothesis to offer, but the reasoning seems appealing but unproven.

---

## Referee Comment (RC2) · Anonymous Referee #2 · 13 Nov 2017

Review on "Microphysical variability of Amazonian deep convective cores observed by CloudSat and simulated by a multi-scale modeling framework" by Dodson et al

In this study the authors use CloudSat data to analyze aspects of the microphysical properties of convective clouds over the Amazon. They show using the DBZ distributions vs. Height space (CFAD) several interesting features of these clouds. They show that when slicing per cloud type, clearer structural information appears that could be interpreted as contribution from specific hydrometeors type. Moreover, comparing the CFAD properties day vs. night and dry vs. wet season they could show shifts in cloud dynamics and microphysics. On the second part of the paper they check if numerical

models can reproduce the CFAD properties. They compare two types of microphysical schemes and show that the two moment can produce results that are closer to the observations but still some of the main features are missing (such as the double arc shown in the observational part).

The first observational part of the paper reads well (less acronyms will make it even better) and it shows quite clearly the differences in the vertical hydrometeors distribution of convective cores, anvils and other clouds. The numerical modeling part is less clear and in my opinion does not stand on the same level of the first part. They start with a long discussion on why the models will fail before showing any results. Then they try to explain the dependency of the model results on updrafts comparing the two microphysical scheme.

Recommendations: (1) I would focus on the observational part making the paper clearer and shorter. The insights gained by the model experiments does not really explain the observations. The message that two moment microphysical scheme produces results that are more similar to the observations is interesting but then why not trying bin microphysics?

(2) One additional analysis that could make the paper clearer and enhance the papers importance is to try to slice the data per aerosol loading. Such analysis should be doable using aerosol information near the observed clouds from the MODIS Aqua. Both the area and the topic of the paper are ideal for such study. Changes in the aerosol loading should directly affect the hydrometeors distribution and therefore should be reflected in the CFAD space. Specifically, in higher aerosol loadings I predict that the convective cores will be taller controlled by stronger updrafts. The onset of precipitation will be delayed but once rain starts it might be stronger. The effect on the ice particle distribution with height would be extremely interesting.

---

## Author Comment (AC1) · 15 Nov 2017

The inclusion of aerosol data to subset the reflectivity observations is an interesting idea, and certainly relevant for Amazonian deep convection. The easiest way for us to do it would be to use CALIPSO monthly mean gridded observations of aerosol optical depth, which are readily available to us. MODIS data would take significantly more time for us to acquire and use. Would CALIPSO suffice as an alternative to MODIS?

---

## Referee Comment (RC3) · Anonymous Referee #2 · 20 Nov 2017

Monthly data might be more sensitive to meteorological differences. Even if comparing the same month for different years the annual variations (droughts, el-nino etc) will modulate the results. Inline with your analysis, I would separate the wet from the dry seasons per one given year and within each season try to combine daily aerosol and CloudSat data. There are few data packages that provide Aqua data in the vicinity of the CloudSat and CALIPSO trajectories.

---

## Author Response (AR1)

**Microphysical variability of Amazonian deep convective cores observed by CloudSat and simulated by a multi-scale modeling framework**

J. Brant Dodson, Patrick C. Taylor, and Mark Branson

*Author Response on ACPD paper acp-2017-864:*

*Thanks to the anonymous referees and editors for reviewing the manuscript. The main changes to the manuscript include a new figure that illustrates the double-arc feature in the CFAD, the inclusion of SPV4 results both with and without diagnosed graupel, and improvements to the writing of the modeling section in order to better mesh the model results with the observations. The point-by-point responses to the referees are found immediately below. Referee comments are displayed in plain text, and responses are shown in italics with indentation. After that is the revised document with tracked changes.*

**Anonymous Referee #1**

Summary:

The authors use the Amazon as a testbed for assessing the internal structure of deep convection observed by CloudSat. Deep convective cores are shown, through a "double arc" structure in CFADS, to be composed of either highly reflective graupel and hail or weakly reflective snow. Cloud structure is contrasted between day/night and wet season/dry season to modest effect. The authors then compare their CloudSat results with those from two SP-CAM runs. These simulations are conducted with different versions of the model which results in the simulations of differing cloud structure between the simulations themselves and between the simulations and CloudSat. The authors report new results, but these are incremental. There are several aspects of the paper that need improvement: 1) the "double arc" is not plainly obvious yet the authors make a point of discussing it at length; 2) the analysis of the simulations seems to lack an obvious direction. I would recommend acceptance if the issues below are addressed.

Primary items:

1) The "double arc" is not especially obvious in any of the panels of Fig. 3. It took me quite a while to fully recognize what structure the authors were talking about and to convince myself that it was not just a result of the contour intervals used. I'm not sure what the remedy is for this, but the double arc structure needs to be made clearer through either some enhancement of the figure, a schematic, or particularly lucid writing.

*We added a figure (now Fig. 3; the old Fig. 3 is now Fig. 4) showing the DCC CFAD, with the double arc structure labeled on the graph.*

2) I don't understand why the authors feel they can ignore graupel in SPV4. The model seems to include graupel to the same degree that it includes any physical species. It seems to be just as much a part of the precipitating ice category as snow.

*Figs. 5-8 are now revised so that they include SPV4 output calculated using a single "precipitating ice" hydrometeor (the old SPV4 results), and SPV4 output calculated with the precipitating ice divided by diagnosed snow and graupel (dubbed SPG4 in the revised paper). The difference between the microphysical parameters used to represent precipitating ice in the radar simulator are much closer to those used for snow than for graupel, so it is reasonable to refer to precipitating ice as snow for simplicity. But as we are now including both variants of hydrometeor calculations in the paper, we can show that the inclusion of graupel in SPG4 results in only a modest change from the results of SPV4. There is still a notable difference between SPG4 and SPV5.*

3) I don't think you have shown sufficient evidence to draw the conclusion you do on Line 285 (even if we all hope that this conclusion is true). Figure 6-9 show only that SPV5 behaves more logically. We do not know how the real world binned variables (reflectivity, SWC, etc) depend on $W_{max}$. And, I'm not sure I agree that the SPV5 CFAD is more like the CloudSat CFAD than the SPV4 CFAD; they share more characteristics with each other than they do with CloudSat. Perhaps you could add the difference between the CloudSat CFAD mean and those from both model runs to Fig. 5d. Or maybe you could compare the variance at each level.

*Perhaps the word "significant" is too strong (as it is ambiguous outside of statistical definitions), but with the modifications to the results section, we do have sufficient evidence to make this conclusion. Bear in mind that Line 285 is the first line of a paragraph, and not meant to be taken out of context. Improving the reflectivity fields in DCCs will not occur as a result of one major change to the model, but as an incremental process through altering several different components of the model. Certainly SPV5 is closer to SPV4 than CloudSat, but it still shows improvement. Improving the microphysics alone won't make the simulation match the observations, but without improving the microphysics, the simulation will not match the observations.*

*I'm not sure how much useful information a rigorous comparison between the CloudSat and SP-CAM CFADs will provide, as it's pretty obvious where the deficiencies in the simulated CFADs lie. Of course we can include one if you think it is vital for the paper. Keep in mind simple methods for quantifying the differences between simulations and observations will not identify the missing double-arc reflectivity structure in the simulations – developing a method to do this might be useful in the future when models improve, but is probably overkill for now.*

Other items:

Line 34: But Fu et al (1990) uses passive sensors. It should be made clear that the second two citations use models.

*The latter two use models, but also observations. Lin et al. use passive sensors as well as TRMM PR, and Zhang et al. use passive radiance and TRMM PR. The point of the citations are to support the claim of the time lag, which is shown in both observations and (some) models.*

Line 42: "There exist lidars" is awkward phrasing.

*exist -> are*

Line 57 to 63: These lines describe the most significant impact this paper will make several pages later.

Line 88: You might want to rephrase "temporal data domain".

*temporal data domain -> time domain for the analysis*

Line 107: Up to this point, you have not explained what scientific purpose you have in focusing on DCC. Is there a reason we expect Amazonian DCC to be especially microphysically variable?

*Much of section 1 is devoted to explaining why we are studying DCCs, and why we chose the Amazon as the place of study. If you find the explanation insufficient, feel free to elaborate on what you think is missing. We don't have a specific a priori reason to expect enhanced microphysical variability in the Amazon above other convectively-active continental regions. We might expect the Amazon to show a range of microphysical characteristics because 1) the variety of convective organizations observed in the Amazon (single cell vs. coastal squall line vs. basin-wide MCC-like clusters), and 2) the variability in aerosol properties of the convective environment. The latter is an interesting possibility, and a possible future topic of investigation, as per RC2's suggestion.*

Line 110: You may want to make it clearer that Fig. 2 does not use your DCC selection.

*It does in the middle row, as the caption indicates. To help the reader in understanding section 3.1, we added the sentence "First, we will look at the frequency of all cloud types, and then subset the clouds into DCCs and anvils."*

Line 122-123: These two sentences seem contradictory.

*The wording is ambiguous. What we meant to say is that in the upper troposphere below 12.5 km, clouds are more frequent during night than day. Above 12.5 km, the opposite is true, suggesting that high clouds are less frequent during the day than night, but occur at higher altitudes. The text has been altered to clarify.*

Figure 3a: What does "Quad" signify?

*All four seasons: wet, dry, wet-to-dry, and dry-to-wet. This has been added to the figure caption.*

Line 150: How do you know that neglecting to separate the data is the reason the double arc has not shown up before? It seems like you could show your selected-CFAD and an all data-CFAD.

*Here is the all-data (i.e. all-sky) CFAD (note the color scale is adjusted to account for the absolute frequency difference):*

[Figure]

*In the CFAD, above 8 km, the low reflectivity arc is much more prominent than the high reflectivity arc. Note than in this particular CFAD, the high reflectivity arc manifests as distinct from the low reflectivity arc, even though it isn't as obviously apparent as the other arc. This is not true for all CFADs shown for convectively-active areas in past research (like those in the cited publications). Clearly, limiting the data to DCC-only makes the two arcs much more distinct, and conversely including all data in a CFAD greatly reduces the contrast. But limiting the data to DCCs only appears to not be the sole factor – other things like the size of the frequency bins, number of observations, etc. are also important. In addition, it may be that the double-arc structure itself is more prominent in some regions than others. We haven't systemically investigated this yet, but we know from our past work that the double-arc structure appears in the tropical oceans (shown by Dodson et al. (2013)) and the contiguous United States (unpublished).*

*So the line in question is mostly justified. We modified the statement to read "...**largely** because the DCCs are not cleanly separated from other cloud types...", because the separation of the data by cloud type is not the sole reason the double-arc structure is prominent in our results.*

Line 167: I'm not sure what the author is implying through the end of this paragraph.

*The comparison of different metrics of convective activity is the topic of a paper of ours currently in preparation. We show that different metrics of convective activity can give different, or even contradictory, answers on analysis methods that depend on convective metrics (in our case the monthly variability in the Amazonian radiative diurnal cycle). It's inspired by the results of Liu et al. (2007). As our paper discussing this is not yet submitted, it is still premature to go into the details in the current paper under review. So to clarify our intentions, we added the sentence "Liu et al. suggest that this metric may be more useful for characterizing convective intensity than cloud top height, a traditional convective metric."*

Figure 3j: The standard deviation of what? Or is it the difference of the standard deviations?

*The standard deviation of the monthly mean day-night contrasts. As this quantity is not relevant for this paper, the standard deviation line has been removed.*

Line 192: Why not show Wet-Dry differences in an additional column (like PM-AM differences are shown in an additional row)? Well, OK, the authors answer this question by the end of the paragraph by punting on seasonal differences. But if they want to do this (which they can), they should probably change the title of section 3.3 to just "Day versus night variability".

*We address the wet-dry season contrast the best we can with the available data. Just because we are forced to punt on the dry season double arc doesn't mean that there is no useful information presented about the wet-dry season contrast. For example, we show that the mean reflectivity profile varies about four times as much in the day-night contrast as wet-dry. To our knowledge, this has not yet been shown by others with CloudSat data.*

Line 220: How was the CRM sampled? Did you sample individual CRM grid columns in a way similar to CloudSat's sampling (i.e. approximately north-south at 1:30am/pm)?

*We sample at 0200 and 1400 LST, which are the model output times closest to the CloudSat overpass times. This is now described in the text.*

Line 226: I'm confused about what the authors mean throughout this paragraph. In SPV4, according to Wang et al (2011), SAM diagnoses all its microphysical species from predicted precipitating and non-precipitating water. Why do the authors feel they can "disregard diagnosed graupel" (and by that I assume they mean when running the radar simulator) and not, say, snow which seems to be diagnosed in the same manner as graupel? It seems like the authors should include graupel in their calculations and change their conclusion that SPV4 underestimates graupel. This would also make the paragraph beginning on Line 230 unnecessary.

*See comment 2 above.*

Line 238: Except SPV4 does "represent" graupel.

*We mean it does not include graupel in the prognostic equations. This paragraph has been rewritten with regard to the inclusion of SPG4 results.*

Figure 5: why use the dry season when the wet season produces the more significant "double-arc"?

*As discussed in the text, the SP-CAM data we have are only available for the dry season. While ideally we would also have wet season data for comparison, it is very unlikely that wet season simulated convection has significantly more realistic reflectivity, let alone a double arc structure. Note despite the double-arc structure being*

*less defined in the dry season observed CFADs than wet season, the CFADs are otherwise much more similar to each other than the simulated CFADs.*

The vertical velocities seem oddly low in magnitude. There is barely any weight to the PDFs above 1m/s (which would usually be considered "convective" in a CRM).

*We've reproduced Fig. 5 using a modified definition of DCCs, where in addition to the criteria in the paper, we also excluded all vertical profiles where $W_{max}$ is less than 1 m/s (here dubbed "ICCs" or (relatively) intense convective cores).*

[Figure]

*The reflectivity values in the mid- to upper troposphere are higher by ~5 dBZ for ICCs in SPV5, but there is no dramatic improvement in the reflectivity, nor is there a double-arc feature. The issue of low vertical velocity is now briefly discussed in the text of Section 4.*

Figure 6: Realistically, I think one has to question the sampling at high Wmax when there are intermediate bins with zero samples. All the panels should probably be cutoff at ~3m/s. What are the different colored lines in panel (c)? Also, the label on the y-axis should probably be something like "fraction" rather than "count".

*It is mentioned in the text that regression values are calculated from 0-2.5 m/s. The full data range are kept for completeness, but we added a vertical line as a reminder of where the cutoff for "useful" data is. The color lines represent night-only and day-only; this is now described in the caption. "Count" is changed to "fraction".*

Line 264: This discussion of the "jump" needs to be clearer.

*Some of the wording has been changed in this paragraph, specifically to highlight how a discontinuous jump would cause a double-arc structure, in other words a bimodal PDF in reflectivity in the upper troposphere.*

Line 270: Or this logical thread about how the double arc structure might arise isn't correct. I don't have a different hypothesis to offer, but the reasoning seems appealing but unproven.

*The other possibility is that the real PDF of $W_{max}$ is bimodal, and the simulated PDFs are not. This is now discussed in the text. There are only three obvious logical possibilities for the discrepancy between the observed and simulated CFADs: the simulated distributions of reflectivity by $W_{max}$ are unrealistic (i.e. Fig. 6a/6b/6c), the simulated PDFs of $W_{max}$ are wrong (i.e. Fig. 6d), or both.*

**Anonymous Referee #2**

In this study the authors use CloudSat data to analyze aspects of the microphysical properties of convective clouds over the Amazon. They show using the DBZ distributions vs. Height space (CFAD) several interesting features of these clouds. They show that when slicing per cloud type, clearer structural information appears that could be interpreted as contribution from specific hydrometeors type. Moreover, comparing the CFAD properties day vs. night and dry vs. wet season they could show shifts in cloud dynamics and microphysics. On the second part of the paper they check if numerical models can reproduce the CFAD properties. They compare two types of microphysical schemes and show that the two moment can produce results that are closer to the observations but still some of the main features are missing (such as the double arc shown in the observational part).

The first observational part of the paper reads well (less acronyms will make it even better) and it shows quite clearly the differences in the vertical hydrometeors distribution of convective cores, anvils and other clouds. The numerical modeling part is less clear and in my opinion does not stand on the same level of the first part. They start with a long discussion on why the models will fail before showing any results. Then they try to explain the dependency of the model results on updrafts comparing the two microphysical scheme.

Recommendations: (1) I would focus on the observational part making the paper clearer and shorter. The insights gained by the model experiments does not really explain the observations. The message that two moment microphysical scheme produces results that are more similar to the observations is interesting but then why not trying bin microphysics?

*As per RC1, the observation section now has an additional figure that more clearly illustrates the double-arc feature in the observed CFADs. This seems to be the aspect of Section 3 that gave many people difficulty.*

*The purpose of the modeling results is not to explain the observations. Their purpose is to illustrate deficiencies in the simulated reflectivity, and suggest possible paths to reconcile simulated and observed reflectivity. This would*

*allow for more robust model/obs. intercomparisons using reflectivity as a metric. We have reorganized and rewritten portions of the modeling study in order to better integrate it with the overall narrative of the paper, specifically to convey this intent, and included a bit more analysis of the effects of microphysics on the simulated reflectivity (specifically the inclusion of diagnosed graupel).*

*Also, it is incorrect to discuss models in terms of failing (in a pass/fail sense), and we are certainly not attempting it in the (paragraph) long discussion at the start of Section 4. It is reasonable to discuss possible source of error in a model as a justification for the following analysis.*

*Finally, improving the microphysics is indeed a message of this paper, and using bin rather than bulk microphysics is a possible route for future models. However, the majority of CRMs today use bulk microphysics because bin microphysics are computationally expensive, and that extra computational power could be used for other purposes, e.g. increasing the spatial resolution. With Moore's law still in effect, the common use of bin microphysics will likely become feasible at some point in the foreseeable future, and so modeling experiments designed to test the benefits of bin over bulk microphysics will be valuable at that time.*

(2) One additional analysis that could make the paper clearer and enhance the papers importance is to try to slice the data per aerosol loading. Such analysis should be doable using aerosol information near the observed clouds from the MODIS Aqua. Both the area and the topic of the paper are ideal for such study. Changes in the aerosol loading should directly affect the hydrometeors distribution and therefore should be reflected in the CFAD space. Specifically, in higher aerosol loadings I predict that the convective cores will be taller controlled by stronger updrafts. The onset of precipitation will be delayed but once rain starts it might be stronger. The effect on the ice particle distribution with height would be extremely interesting.

*As we mentioned in the interactive comments, this is a very good idea, and something that is reasonably doable. However, we decided that this paper is not the best place for such an analysis, for several reasons.*

*1) In general, such an analysis is beyond the scope of this paper.*

*2) We're concerned that if we attempt to tack on an analysis of aerosol data to this paper, it would likely come off as a rushed, half-baked attempt to cram more into a paper that should have a complete narrative with just the observations and SP-CAM material. The effect of aerosols on convection is an important question, and one that deserves its own paper, and associated narrative.*

*3) Aerosols are complex. It probably would not be right to choose just a single variable for aerosols, such as total optical depth, but look at the effect of different aerosol properties (chemical species, size, vertical distribution) on convection. A rushed analysis would not allow sufficient time to investigate these different properties.*

*4) Unfortunately, the reviewer did not answer the question of whether CALIPSO data would be a useful alternative to MODIS data, or give particular motivation for using MODIS is particular. If we leave the aerosol analysis to a separate paper, we would have time to implement both datasets, and determine whether the results are dependent on the choice of dataset.*

*For these reasons, we will not include aerosols in this paper. However, we are strongly considering it as the topic for a future paper, and so we thank the reviewer for the suggestion.*

[revised manuscript text omitted]

SPv4 DCC refl vs. Wmax

SPV5 DCC refl vs. Wmax

SPV4 and SPV5 Wmax Histogram

625

[Figure]

**Figure 68.** (a) Mean vertical profiles of simulated DCC radar reflectivity in SPV4 sorted by maximum updraft velocity. The vertical lines at 2.5 m s$^{-1}$ represent the maximum cut off W$_{max}$ value for statistical calculations. Vertical black stripes are updraft values with no DCC occurrences.

630 (b) Same as (a), but for SPG4 data (i.e., SPV4 with diagnosed graupel included in the reflectivity calculation).

(c) Same as (a), but for SPV5 data.

(ed) PDFs of maximum DCC updraft velocity for (solid) SPV4 and (dashed) SPV5. The black line represents data from both 0200 and 1400 LST, the blue line from 0200 LST only (night), and the red line from 1400 LST only (day).

[Figure]

[Figure]

635

**Figure 7. Same as Fig. 6Figure 9. Same as Fig. 8, but for snow water content (GWC).SWC). Because SPV4 does not distinguish between snow and graupel, all precipitating ice are depicted as SWC.**

[Figure]

[Figure]

640        Figure 8̶10. Same as Fig. 6̶8, but for graupel water content (GWC). Note the absence of GWC for SPV4.

[Figure]

[Figure]

---

## Author Response (AR2)

*Co-Editor Decision: Publish subject to minor revisions (review by editor) (23 Feb 2018) by Philip Stier*

*Comments to the Author:*

*Dear Authors*

*Thank you for providing the revised manuscript "Microphysical variability of Amazonian deep convective cores observed by CloudSat and simulated by a multi-scale modeling framework" as well as your detailed responses to the issues raised during the reviews.*

*Based on the reviewer's comments and my own assessment I have concluded to accept your manuscript for publication in Atmospheric Chemistry and Physics subject to minor revisions.*

*Please address the issues listed below in your revised manuscript.*

*Kind regards,*

*Philip Stier*

*Co-Editor*

*Atmospheric Chemistry and Physics*

*List of issues to address:*

*1) Figure 3 (and more generally): please avoid introduction of unnecessary acronyms, such as Quad - usage of e.g. "all seasons" is much easier to understand.*

"Quad" has been removed, along with a couple other modifications (e.g. "ANV" spelled out as "anvil")

*2) The reviews raised questions about the origin of the double arc structure. Is this something that could be investigated from microphysical signatures in dual-polarisation radar data, which should have been available for GoAmazon?*

We've been in contact with other researchers using GOAmazon data to examine the properties of convection, such as Scott Giangrande, to figure out how to combine information from CloudSat with field experiment observations. This could be either directly, by matching GOAmazon radar scans with CloudSat overpasses, or indirectly, by comparing the statistics of CloudSat observations with those of the surface instruments. It might be possible to use the horizontally-scanning W-band radar to observe the double-arc structure in convection occurring during the field experiment. Dual polarization radars would be able to detect the larger hydrometeors (likely) responsible for the double-arc structure - hail, graupel, and snow/aggregates - and give us information about the hydrometeor phase. That could be combined with the W-band radar data to further understand the properties of the double arc structure. I would think the main limitation would be sample size of events observed by GOAmazon, particularly during the dry season. It's an interesting idea.

*3) Line 157: this should be Figure 4b?*

fixed

*3) Line 192: this should be Figure 5?*

45  fixed

*4) Section 4: this should be merged with the data and methodologies section, it does not fit in the results section (only the results in 4.1... do)*

50  The sections have been rearranged as requested.

*5) Include the effective resolution / grid-spacing of the SP-CAM model*

4 km horizontal grid spacing, 100 m – 1 km vertical grid spacing (varying by altitude). Now included.

55

*6) Line 275, Figure 8-11: You mix the terminology for the same results: CRM/SP-CAM/SPV4. Please simplify your terminology and use consistently throughout.*

We found that most of the confusion involves Fig. 6-7, where the terminology "SP-CAM version X" is used in the captions
60  instead of "SPXX". We changed the captions to the "SPXX" terms for consistency. Otherwise, we don't see where the
terminology causes serious confusion. "SP-CAM" refers to the whole family of super-parameterized CAMs, "CRM" refers to the
cloud-resolving model-component of SP-CAM, and "SPV4/SPG4/SPV5" refer to specific versions of the SP-CAM family. Also,
related to this, we changed the "why the missing double arc" discussion from being SPV5-exclusive to SP-CAM in general, as
we are arguing that there are more issues than just the missing graupel in SPV4.
65  One other change we made is that we thought that the use of acronyms and initialisms in general caused too much of an alphabet
soup, so we removed the use of CTH and ETH to lessen their prevalence.

*7) Make figures consistent (don't use grey background in a but not in b,.... ) and ensure good quality.*

70  Which figure in particular is the issue? We don't see any figures in the current draft where this is an issue. It was a problem in
previous drafts, but that was fixed. The only cases where panels of a figure switch background colors are when different panels
show different variable types, and thus use different color palettes.

[revised manuscript text omitted]